# IDEA: INVARIANT CAUSAL DEFENSE FOR GRAPH ADVERSARIAL ROBUSTNESS

## ABSTRACT

Despite the success of graph neural networks (GNNs), their vulnerability to adversarial attacks poses tremendous challenges for practical applications. Existing defense methods suffer from severe performance decline under some unknown attacks, due to either limited observed adversarial examples (adversarial training) or pre-defined heuristics (graph purification or robust aggregation). To address these limitations, we analyze the causalities in graph adversarial attacks and conclude that causal features are desirable to achieve graph adversarial robustness, owing to their determinedness for labels and invariance across attacks. To learn these causal features, we innovatively propose an *Invariant causal DEfense method against adversarial Attacks* (IDEA). We derive node-based and structure-based invariance objectives from an information-theoretic perspective. IDEA is provably a causally invariant defense across various attacks. Extensive experiments demonstrate that IDEA significantly outperforms all baselines under both poisoning and evasion attacks on five benchmark datasets, highlighting its strong and invariant predictability. The implementation of IDEA is available at `https://anonymous.4open.science/r/IDEA_repo-666B`.

## 1 INTRODUCTION

Graph neural networks (GNNs) have achieved immense success in numerous tasks and applications, including node classification (Kipf & Welling, 2017; Veličković et al., 2018; Xu et al., 2019b;a), cascade prediction (Cao et al., 2020), recommendation (Fan et al., 2019), and fraud detection (Ma et al., 2021; Cheng et al., 2022). However, GNNs have been found to be vulnerable to adversarial attacks (Dai et al., 2018; Zügner et al., 2018; Bojchevski & Günnemann, 2019), i.e., imperceptible perturbations on graph data can easily mislead GNNs into misprediction (Jin et al., 2020a; Chen et al., 2020; Zügner & Günnemann, 2019b; Lin et al., 2020; Zügner et al., 2018). For example, in credit scoring, attackers add fake connections with high-credit customers to deceive GNN (Jin et al., 2020a), leading to loan fraud and severe economic losses. This vulnerability poses significant security risks, hindering the deployment of GNNs in real-world scenarios. Therefore, defending against adversarial attacks is crucial for practical utilization of GNNs, and has attracted substantial research interests.

Existing defense methods (Jin et al., 2020a) mainly include graph purification, robust aggregation, and adversarial training, showing effectiveness against specific attacks. Graph purification purifies adversarial perturbations based on low rank (Jin et al., 2020b; Entezari et al., 2020), local smoothness (Li et al., 2022c), or sparsity (Jin et al., 2020b). Robust aggregation assigns high weight to edges, also based on local smoothness (Jin et al., 2021; Zhang & Zitnik, 2020). Adversarial training employs a min-max optimization scheme (Kong et al., 2022; Li et al., 2022b), iteratively generating adversarial examples to maximize the loss and update GNN to minimize the loss on these examples.

However, these approaches all have limitations that prevent broad protection across various attacks. Graph purification and robust aggregation both rely on specific heuristic priors, but these priors may be ineffective for some attacks (Chen et al., 2022b; Tao et al., 2023a), causing the methods to fail. As shown in Figure 1 (a), graph purification methods including ProGNN (Jin et al., 2020b), STABLE (Li et al., 2022c), and GARNET (Deng et al., 2022) perform well under MetaAttack (Zügner et al., 2018) (light green and dark green), however, they suffer severe performance degradation when faced with G-NIA (Tao et al., 2021b) (red). On the other hand, robust aggregation methods including SimPGCN (Jin et al., 2021), Elastic(Liu et al., 2021), and Soft-Median (Geisler et al., 2021) also

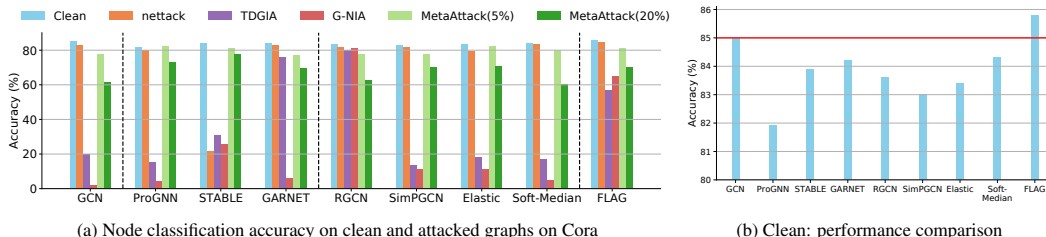

(a) Node classification accuracy on clean and attacked graphs on Cora

(b) Clean: performance comparison

Figure 1: Limitation of existing methods: Defenses suffer performance degradation under various attacks and on clean graph. 5% and 20% denote perturbation rates of MetaAttack.

exhibit weaknesses against TDGIA (Zou et al., 2021) and G-NIA. Moreover, modifying graph structure (ProGNN) or adding noise (RGCN) can even degrade clean graph performance, as shown in Figure 1(b) using GCN as a reference. As for adversarial training, its effectiveness is limited to the observed examples (Bojchevski & Günnemann, 2019). As shown in Figure 1 (a), adversarial training FLAG (Kong et al., 2022) also exhibits unsatisfactory performance under TDGIA. Due to limited space, we discuss further literature in Appendix A.

To address the above limitations issues, we innovatively propose an invariant causal defense perspective. Specifically, we first design an interaction causal model (Zhang et al., 2022a) to capture causalities between nodes in graph adversarial attacks, addressing non-IID characteristics of graph data. Then we find that the causal features exhibit good properties: (1) Causal features determine labels, implying their **strong predictability for labels**; (2) The causalities between causal features and labels are invariant across attacks, indicating their **invariant predictability across attacks**. We conclude that causal features are advantageous to achieve graph adversarial robustness.

To learn these causal features, we propose an *Invariant causal DEfense method against adversarial Attacks, namely IDEA*. By analyzing distinct characteristics of causal features, we derive node-based and structure-based invariance objectives from an information-theoretic perspective. Node-based invariance objective minimizes the conditional mutual information between label and attack given causal feature, based on that the causality between causal feature and label remains unchanged across attacks. While structure-based invariance objective is specially designed considering graph structure. IDEA is proved to be a causally invariant defense, under the linear causal assumptions. Extensive experiments demonstrate that IDEA attains state-of-the-art defense performance under all five attacks on all five datasets. This emphasizes that IDEA's strong and invariant predictability across attacks.

The primary contributions of this paper can be mainly summarized as:

1. *New perspective:* We introduce an innovative invariant causal defense perspective and design an interaction causal model to capture the causalities in graph adversarial attack, offering a novel perspective on graph adversarial field.

2. *Novel methodology:* We propose IDEA method to learn causal features to achieve graph adversarial robustness. We design two invariance objectives to learn causal features by modeling and analyzing the causalities in graph adversarial attacks.

3. *Experimental evaluation:* Comprehensive experiments on five benchmarks demonstrate that IDEA significantly outperforms all baselines against both evasion and poisoning attacks, highlighting the strong and invariant predictability of IDEA.

## 2 PRELIMINARY

In this section, we introduce the widely-used node classification task and graph neural networks. We also introduce the goal of graph adversarial attack and graph adversarial robustness.

**GNN for Node Classification.** Given an attributed graph $G = (\mathcal{V}, \mathcal{E}, X)$, we denote $\mathcal{V} = \{1, 2, ..., n\}$ as node set, $\mathcal{E} \subseteq \mathcal{V} \times \mathcal{V}$ as edge set, and $X \in \mathbb{R}^{n \times d}$ as the attribute matrix with $d$-dimensional attributes. The class set $\mathcal{K}$ contains $K = |\mathcal{K}|$ classes. The goal is to assign labels for nodes based on the node attributes and network structure by learning a GNN classifier $f_\theta$ (Jin et al., 2020b;a). The objective is: $\min_\theta \sum_{i \in \mathcal{V}_{\text{train}}} [L(f_\theta(G)_i, Y_i)]$, where $Y_i$ denotes the ground-truth label of node $i$.

**Graph adversarial attacks.** The graph adversarial attack aims to find a perturbed graph $\hat{G}$ that maximizes the loss of GNN model (Jin et al., 2020a):

$$\max_{\hat{G} \in \mathcal{B}(G)} \sum_{i \in \mathcal{V}} [L(f_{\theta^*}(\hat{G})_i, Y_i)] \qquad s.t., \quad \theta^* = \arg\min_{\theta} \sum_{i \in \mathcal{V}} [L(f_\theta(G_{\text{train}})_i, Y_i)]. \tag{1}$$

Here, $\hat{G}$ is the perturbed graph chosen from the admissible perturbed graph set $\mathcal{B}(G)$, where the perturbing nodes, edges, and node attributes should not exceed the corresponding budget (Jin et al., 2020a). $G_{\text{train}} = G$ in evasion attacks, and $G_{\text{train}} = \hat{G}$ in poisoning attacks.

**Graph adversarial robustness.** Defense methods aim to improve graph adversarial robustness, defending against any adversarial attack. The goal can be formulated as:

$$\min_{\theta} \max_{\hat{G} \in \mathcal{B}(G)} \sum_{i \in \mathcal{V}} [L(f_\theta(\hat{G})_i, Y_i)]. \tag{2}$$

Existing defense methods suffer performance degradation under various attacks or on clean graphs. Adversarial training (Kong et al., 2022) generalizes poorly to unseen adversarial attacks. While graph purification and robust aggregation are designed based on specific heuristic priors, such as local smoothness (Wu et al., 2019; Jin et al., 2020b; Zhang & Zitnik, 2020; Li et al., 2022c; Jin et al., 2021) and low rank (Jin et al., 2020b; Entezari et al., 2020). They are only effective when attacks satisfy these priors. Hence, there is an urgent need to design a defense method that performs well both on clean graphs and across various attacks.

## 3 METHODOLOGY

We first model the causalities between causal features and other variables in graph adversarial attacks. Based on this causal analysis, we propose an Invariant causal DEfense method against Attacks (IDEA) method to learn causal features.

### 3.1 INTERACTION CAUSAL MODEL

To model the non-IID characteristics in graph data, namely the interactions (e.g. edges) between samples (e.g. nodes), we design an interaction causal model with explicit variables [1] to capture the causality between different samples (Zhang et al., 2022a) under graph adversarial attacks.

Figure 2 (left) illustrate an example involving two connected nodes $i$ and $k$. We inspect the causal relationships among variables: input data $G_i$ (node $i$'s ego-network), label $Y_i$, causal feature $C_i$, perturbation $T_i$, attack domain $D_i$, and those variables of neighbor node $k$.

We introduce the latent causal feature $C_i$ as an abstraction that causes both input ego-network $G_i$ and label $Y_i$. For example, in credit scoring, $C_i$ represents the financial situation, which determines both $G_i$ (including personal attributes and friendships) and credit score $Y_i$. Besides, the causal feature $C_i$ influences neighbor $G_k$ due to network structure, aligning with GNN studies (Kipf & Welling, 2017; Veličković et al., 2018). We model graph adversarial attack with perturbation $T_i$ and attack domain $D_i$ which is a latent factor that determines $T_i$, as shown in Figure 2 (left). Attack domain $D_i$ refers to attack categories based on their characteristics, such as attack type or attack strength. Here, $D_i$ and $T_i$ are considered as non-causal features $N_i$ associated to attack, and we strive to exclude their influence. Perturbation $T_i$ may impact the neighbor ego-network $G_k$ due to edges between nodes.

We analyze these causalities and find that: (i) Causal feature $C$ determines label $Y$, indicating causal feature's **strong predictability** for label; (ii) The $C - Y$ causality remains unchanged across attack domains, indicating the causal feature maintains **invariant predictability** across attack domains. These properties make causal features beneficial in enhancing graph adversarial robustness. Specifically, strong predictability enables good performance on clean graphs, while invariant predictability maintains performance under attacks. Meanwhile, the impact of attacks including $D_i$ and $T_i$ should be eliminated. Based on the above intuition, we aim to design a method to learn causal feature $C$ and reduce the influence of non-causal feature to defend against attacks.

---

[1]Note that the explicit variable $X_i$ refers to the event of one specific sample $i$, and the generic variable $X$ is the event of all samples.

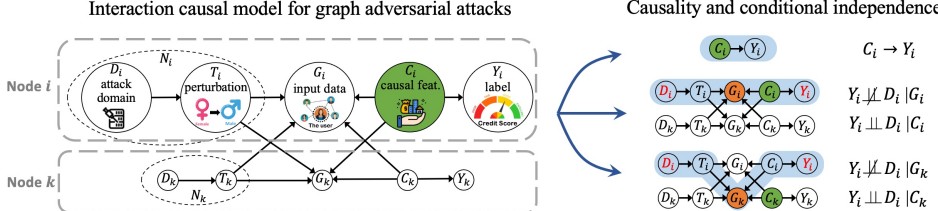

Figure 2: **Left**: Interaction causal model. **Right**: Causality and conditional independences.

## 3.2 IDEA: INVARIANT CAUSAL DEFENSE METHOD AGAINST ADVERSARIAL ATTACK

We propose IDEA to learn causal features. Our approach involves designing invariance objectives based on the distinctive properties of causal features and approximating losses accordingly.

### 3.2.1 INVARIANCE OBJECTIVE

To learn causal features, we design invariance objectives by analyzing the causality and conditional independences with the $d$-separation (Pearl, 2010; 2009) in Figure 2 (right). The observations are:

1. $C_i \to Y_i$: Causal feature $C$ determines label $Y$. (first path in Figure 2 (right)).
2. $Y_i \not\perp\!\!\!\perp D_i \mid G_i$: $Y$ and attack domain $D$ are associated given $G$ (second path in Figure 2 (right)), since $G$ is a collider[2] between $Y$ and $D$.
3. $Y_i \perp\!\!\!\perp D_i \mid C_i$: The $C - Y$ causality remains unchanged across attack domain $D$, i.e., $C$ has invariant predictability for $Y$ across various attack domains.
4. $Y_i \not\perp\!\!\!\perp D_i \mid G_k$: $Y_i$ and $D_i$ are associated given the neighbor's ego-network $G_k$ (third path in Figure 2(right)), since $G_k$ is a collider of $Y_i$ and $D_i$, where $G_k$ is also influenced by attacks.
5. $Y_i \perp\!\!\!\perp D_i \mid C_k$: Given neighbor's $C_k$, $Y_i$ and $D_i$ are still independent.

Based on these observations, we analyze the characteristics of $C$ and propose three goals from the perspective of mutual information $I$ to learn causal feature $C$. Let $\Phi$ represent the feature encoder.

- **Predictive goal**: $\max_\Phi I\left(\Phi(G), Y\right)$ to guide $\Phi(G)$ to have strong predictability for $Y$ in (1).
- **Node-based Invariance goal**: $\min_\Phi I\left(Y, D \mid \Phi(G)\right)$. By comparing the conditional independences regarding $G_i$ in (2) and $C_i$ in (3), we propose this goal to guide $\Phi(G)$ learning $C$ and excluding the influence of attack, to obtain invariant predictability across different attack domains.
- **Structural-based Invariance goal**: $\min_\Phi I\left(Y, D \mid \Phi(G)_\mathcal{N}\right)$. Through comparing $G_k$ in (4) and $C_k$ in (5), we propose this goal to guide $\Phi(G)_\mathcal{N}$ learning the causal feature for neighbor $\mathcal{N}$.

To sum up, the objective can be formulated as:

$$\max_\Phi \quad I\left(\Phi(\hat{G}^*), Y\right) - \left[I\left(Y, D \mid \Phi(\hat{G}^*)\right) + I\left(Y, D \mid \Phi(\hat{G}^*)_\mathcal{N}\right)\right]$$
$$\text{s.t.} \quad \hat{G}^* = \arg\min_{\hat{G} \in \mathcal{B}(G)} I\left(\Phi(\hat{G}), Y\right), \tag{3}$$

Here, attack domain $D$ is used to expose the difference of attack influence on feature, mitigate thereby learning the causal features that are invariant across attacks. The objective guides IDEA to learn causal feature with strong predictability and invariant predictability across attack domains, as well as exclude the impact of attacks. The capability of this objective relies on the diversity of attack domain $D$ (Section 3.2.3). However, two challenges persist in solving Eq. 3: i) The objective is not directly optimizable since estimating mutual information of high-dimensional variables is difficult. ii) It is unknown that how to design attack domain $D$. Intuitively, diverse $D$ can expose the difference of attack influence on features and promote learning inviarant causal feature.

To address the challenge i), section 3.2.2 presents the loss approximations for our proposed objectives. To tackle the challenge ii), section 3.2.3 introduces a domain learner to learn the attack domain.

---

[2] A collider is causally influenced by two variables and blocks the association between the variables that influence it. Conditioning on a collider opens the path between its causal parents (Pearl, 2010; 2009). In our case, $C_i$ and $T_i$ are associated conditioned on $G_i$, making $Y_i$ and $D_i$ associated, conditioned on $G_i$.

### 3.2.2 LOSS APPROXIMATION

Let $Z = \Phi(\hat{G}^*)$ denote the representation to learn $C$. The predictive goal is $I(Z, Y) = \mathbb{E}_{p(y,z)} \log \frac{p(y|z)}{p(y)}$. Unfortunately, it is challenging to directly compute the distribution $p(y|z)$. To overcome this, we introduce $q(y|z)$ as a variational approximation of $p(y|z)$. Similar to (Alemi et al., 2017; Li et al., 2022a), we derive a lower bound of predictive goal: $\mathbb{E}_{p(x,y)}\mathbb{E}_{p(z|x)} \log q(y \mid z)$, where $x$ is the input ego-network. Consequently, we can maximize $I(Z, Y)$ by maximizing the lower bound.

The lower bound involves two distributions, $p(z|x)$ and $q(y|z)$ that need to be solved. For $p(z|x)$, we employ neural network $h$ as a realization of $\Phi$ to learn representation. We assume a Gaussian distribution $p(z \mid x) = \mathcal{N}\left(z \mid h^\mu(x), h^\Sigma(x)\right)$, where $h^\mu$ and $h^\Sigma$ output the mean and covariance matrix, respectively (Li et al., 2022a; Alemi et al., 2017). Subsequently, we leverage a re-parameterization technique (Kingma & Welling, 2014) to tackle non-differentiability: $z = h^\mu(x) + \epsilon h^\Sigma(x)$, where $\epsilon$ is a standard Gaussian random variable. In general, encoder $h$ contains GNN and re-parameterization, outputting $z = h(x, \epsilon)$. For $q(y|z)$, we use a neural network $g$ as classifier to learn the variation distribution $q(y|z)$. With $p(z|x)$ and $q(y|z)$, we obtain the predictive loss $\mathcal{L}_\mathcal{P}$:

$$\min_{g,h} \mathcal{L}_\mathcal{P}\left(g, h, \hat{G}^*\right) = \min_{g,h} \sum_{i \in \mathcal{V}_l} L(g(h(\hat{G}^*)_i), Y_i). \tag{4}$$

For the node-based invariance goal, the conditional mutual information is defined as:

$$I(Y, D|Z) = \mathbb{E}_{p(z)} \left[ \mathbb{E}_{p(y,d|z)} \left[ \log \frac{p(y, d \mid z)}{p(d \mid z)p(y \mid z)} \right] \right]$$

$$= \mathbb{E}_{p(z)} \left[ \mathbb{E}_{p(y,d|z)} [\log p(y \mid z, d) - \log p(y \mid z)] \right]. \tag{5}$$

To approximate $p(y \mid z, d)$, $p(y \mid z)$, we also employ two variational distributions $q_d(y \mid z, d)$ and $q(y \mid z)$. This allows us to obtain an estimation of $I(Y, D|Z)$:

$$\hat{I}(Y, D|Z) = \mathbb{E}_{p(z)} \left[ \mathbb{E}_{p(y,d|z)} [\log q_d(y \mid z, d) - \log q(y \mid z)] \right]. \tag{6}$$

Similar to CLUB (Cheng et al., 2020), we minimize KL-divergence $\mathbb{E}_{p(z,d)} KL[p(y \mid z, d) \| q_d(y \mid z, d)]$ to make $\hat{I}(Y, D|Z)$ as an upper bound on $I(Y, D|Z)$. We prove that minimizing both estimation $\hat{I}(Y, D|Z)$ and KL-divergence $\mathbb{E}_{p(z,d)} KL[p(y \mid z, d) \| q_d(y \mid z, d)]$ minimizes our goal $I(Y, D|Z)$.

**Proposition 1.** *The node-based invariance goal $I(Y, D|Z)$ reaches its minimum value if the following two quantities are minimized: $\mathbb{E}_{p(z,d)} KL[p(y \mid z, d) \| q_d(y \mid z, d)]$ and $\hat{I}(Y, D|Z)$.*

The proof is in Appendix B.1. Then we use a neural network $g_d$ as classifier to learn $q_d(y \mid z, d)$. We optimize $\mathbb{E}_{p(z,d)} KL[p(y \mid z, d) \| q_d(y \mid z, d)]$ by minimizing $\sum_{i \in \mathcal{V}} L\left(g_d(h(\hat{G}^*)_i, D_i), Y_i\right)$, and optimize $\hat{I}(Y, D|Z)$ by minimizing $\sum_{i \in \mathcal{V}} \left[ L\left(g(h(\hat{G}^*)_i), Y_i\right) - L(g_d(h(\hat{G}^*)_i, D_i), Y_i) \right]$. The node-based invariance loss $\mathcal{L}_\mathcal{I}$:

$$\min_{g,g_d,h} \mathcal{L}_\mathcal{I}\left(g, g_d, h, \hat{G}^*, D\right)$$

$$= \min_{g,g_d,h} \sum_{i \in \mathcal{V}} L\left(g_d\big(h(\hat{G}^*)_i, D_i\big), Y_i\right) + \alpha \left[ L\Big(g\big(h(\hat{G}^*)_i\big), Y_i\Big) - L\Big(g_d\big(h(\hat{G}^*)_i, D_i\big), Y_i\Big) \right], \tag{7}$$

where coefficient $\alpha$ is a hyper-parameter to balance the two terms.

The structural-based invariance goal aims to learn the causal feature for the neighbor. Similar to the above, this goal can be achieved by optimizing the structure-based invariance loss $\mathcal{L}_\mathcal{E}$:

$$\min_{g,g_d,h} \mathcal{L}_\mathcal{E}\left(g, g_d, h, \hat{G}^* D\right)$$

$$= \min_{g,g_d,h} \sum_{i \in \mathcal{V}, k \sim \mathcal{N}_i} L\left(g_d\big(h(\hat{G}^*)_k, D_i\big), Y_i\right) + \alpha \left[ L\Big(g\big(h(\hat{G}^*)_k\big), Y_i\Big) - L\Big(g_d\big(h(\hat{G}^*)_k, D_i\big), Y_i\Big) \right], \tag{8}$$

where $k$ is the sampled node from the neighbors $\mathcal{N}_i$ of node $i$.

In summary, the loss function consists of predictive loss and two newly proposed invariance losses:

$$\min_{g,g_d,h} \mathcal{L}_\mathcal{P}\left(g, h, \hat{G}^*\right) + \mathcal{L}_\mathcal{I}\left(g, g_d, h, \hat{G}^*, D\right) + \mathcal{L}_\mathcal{E}\left(g, g_d, h, \hat{G}^*, D\right)$$

$$\text{s.t. } \hat{G}^* = \arg \max_{\hat{G} \in \mathcal{B}(G)} \mathcal{L}_\mathcal{P}\left(g, h, \hat{G}\right). \tag{9}$$

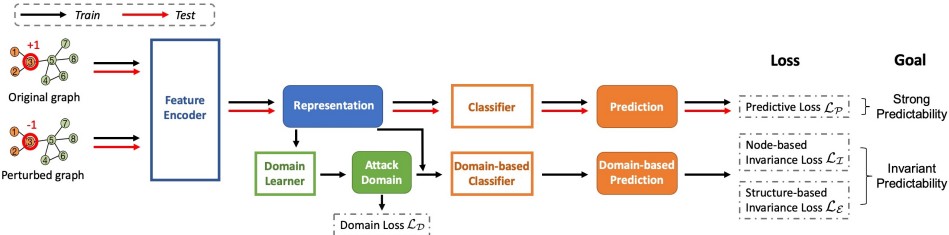

Figure 3: Overall architecture of our IDEA method. IDEA contains feature encoder, classifier, domain-based classifier, and domain learner. The black arrows denote the workflow of IDEA during training, while the red arrows denote how IDEA predicts in the test phase.

### 3.2.3 DOMAIN CONSTRUCTION

How to design attack domain $D$ remains critical. Straightforward ways such as categorizing by attack type or strength can yield very few, non-diverse domains. Intuitively, attack domains should be both sufficiently numerous (Rosenfeld et al., 2021; Arjovsky et al., 2019) and distinct from each other (Creager et al., 2021; Arjovsky et al., 2019) to reveal the various effects of attacks. To this end, we leverage a neural network $s$ as domain learner to learn attack domain. Here, $s$ allows for the adjustable number of domains. We ensure the domain diversity by minimizing the co-linearity between the samples from different domains. We adopt Pearson correlation coefficient (**PCCs**) to measure of linear correlation between two sets of data. The loss function $\mathcal{L}_\mathcal{D}$:

$$\min_s \mathcal{L}_\mathcal{D}\left(s, h\right) = \min_s \sum_{D, D' \in \mathcal{D}, D \neq D'} \mathbf{PCCs}\left(r^D, \rho(\hat{G})^{D'}\right)$$

$$r^D = \mathbb{E}_{i \in \mathcal{V}^D}\left[h(\hat{G}^*)_i\left(g(h(\hat{G}^*)_i) - Y_i\right)\right], \mathcal{V}^D = \left\{i|(s(h(\hat{G}^*)_i) = D\right\},$$

(10)

where attack domain $D$ is in the form of one-hot vector to categorize adversarial samples, $\mathcal{D}$ is the attack domain set, $\mathcal{V}^D$ denotes nodes assigned to domain $D$ by learner $s$, $r^D$ denotes the representation of $\mathcal{V}^D$. The form of $r^D$ aids in proving IDEA achieving adversarial robustness (Proposition 2).

### 3.2.4 OVERALL FRAMEWORK

According to the above analysis, the overall loss function of IDEA is formulated as:

$$\min_{g, g_d, h} \mathcal{L}_\mathcal{P}\left(g, h, \hat{G}^*\right) + \mathcal{L}_\mathcal{I}\left(g, g_d, h, \hat{G}^*, s^*\right) + \mathcal{L}_\mathcal{E}\left(g, g_d, h, \hat{G}^*, s^*\right)]$$

$$\text{s.t. } \hat{G}^* = \arg\max_{\hat{G} \in \mathcal{B}(G)} \mathcal{L}_\mathcal{P}\left(g, h, \hat{G}\right), \quad s^* = \arg\min_s \mathcal{L}_\mathcal{D}\left(s, h\right).$$

(11)

The overall architecture of IDEA is illustrated in Figure 3. The IDEA model consists of four parts: an encoder $h$ to learn the node representation, i.e., causal feature; a classifier $g$ for final classification; a domain-based classifier $g_d$ for invariance goals; and a domain learner $s$ to provide the partition of attack domain. We also provide the algorithm in Appendix C.

Through theoretical analysis in Proposition 2, IDEA produces causally invariant defenders under the linear assumption of causal relationship (Arjovsky et al., 2019), enabling graph adversarial robustness.

**Proposition 2.** *Let $Y = C\gamma + \epsilon$ where $\gamma$ is the causal mechanism, $\epsilon \sim \mathcal{N}(0, \sigma^2)$ is Gaussian noise. Let $\rho(\hat{G}) = \psi(C, N)$ where $\psi$ is the mapping from causal feature $C$ and non-causal feature $N$ to graph representation $\rho(\hat{G})$, and $\rho$ is a powerful graph representation extractor can extract all information from $\hat{G}$. Encoder $\Phi$ comprises $\rho$ and a learner $\phi$ with parameter $\Theta_\phi$ to learn $C$. Suppose a function $\tilde{\psi}$ satisfying $\tilde{\psi}(\rho(\hat{G})) = C$, with parameters $\Theta_{\tilde{\psi}}$. Assume the rank of $\Theta_\phi$ is $r$. Let $\Theta_{\tilde{\psi}}^\top\Theta_\gamma$ and $\Theta_\phi^\top\Theta_\omega$ be the parameter of the ground truth defender and learned defender. If $\Theta_\phi^\top\Theta_\omega$ satisfies the following conditions in training attack domain set $\mathcal{D}_{tr}$:*
*(1) Eq. 3, $I\left(\Phi(\hat{G}), Y\right) - \left[I\left(Y, D \mid \Phi(\hat{G})\right) + I\left(Y, D \mid \Phi(\hat{G})_\mathcal{N}\right)\right]$ is maximized,*
*(2) $\left\{\mathbb{E}_{\rho(\hat{G}^D)}\left[\rho(\hat{G}^D)\rho(\hat{G}^D)^\top\right]\left(\Theta_\phi^\top\Theta_\omega - \Theta_{\tilde{\psi}}^\top\Theta_\gamma\right)\right\}_{D \in \mathcal{D}_{tr}}$ is linearly independent and $\dim\left(\text{span}\left(\left\{\mathbb{E}_{\rho(\hat{G})_i}\left[\rho(\hat{G})_i\rho(\hat{G})_i^\top\right]\left(\Theta_\phi^\top\Theta_\omega - \Theta_{\tilde{\psi}}^\top\Theta_\gamma\right)\right\}_{i \in \mathcal{V}}\right)\right) > \dim(\phi) - r$,*
*then $\Theta_\phi^\top\Theta_\omega = \Theta_{\tilde{\psi}}^\top\Theta_\gamma$ is causal invariant defender for all attack domain set $\mathcal{D}_{all}$.*

Table 1: Accuracy(%) of targets under evasion attacks. The **best** and second-best are highlighted. Parentheses denote IDEA's relative increase compared to second-best. "-" for out-of-memory (OOM).

| Dataset | Attack | GCN | GAT | ProGNN | STABLE | GARNET | RGCN | SimpGCN | Elastic | Soft-Median | FLAG | IDEA |
|---|---|---|---|---|---|---|---|---|---|---|---|---|
| Cora | Clean | 85.0 ± 0.5 | 84.6 ± 0.8 | 81.9 ± 1.2 | 83.9 ± 0.6 | 84.2±0.8 | 83.6 ± 0.7 | 83.0 ± 1.2 | 83.4 ± 1.9 | 84.3±0.9 | 85.8 ± 0.6 | **88.4 ± 0.6** (↑ 3.0%) |
| | nettack | 83.0 ± 0.5 | 81.7 ± 0.7 | 79.9 ± 1.1 | 21.5 ± 4.8 | 83.0±1.0 | 81.7 ± 0.6 | 82.0 ± 1.2 | 79.4 ± 1.7 | 83.5±0.8 | 84.8 ± 0.6 | **85.4 ± 0.7** (↑ 0.8%) |
| | PGD | 44.2 ± 3.4 | 26.7 ± 7.6 | 19.6 ± 2.2 | 32.2 ± 0.2 | 81.4±1.6 | 80.5 ± 0.4 | 9.0 ± 2.2 | 29.0 ± 5.5 | 19.5±2.3 | 60.2 ± 2.4 | **83.6 ± 2.1** (↑ 2.6%) |
| | TDGIA | 20.2 ± 2.3 | 33.7 ± 14.9 | 15.4 ± 1.7 | 30.9 ± 2.4 | 76.1±2.9 | 79.9 ± 0.9 | 13.5 ± 1.2 | 18.0 ± 1.2 | 16.8±0.4 | 57.2 ± 3.0 | **81.2 ± 2.5** (↑ 1.6%) |
| | G-NIA | 2.3 ± 0.5 | 5.2 ± 3.1 | 4.2 ± 0.8 | 25.8 ± 10.1 | 6.1±0.9 | 81.3 ± 0.9 | 11.5 ± 8.1 | 11.2 ± 3.7 | 4.8±0.8 | 64.8 ± 2.0 | **85.3 ± 1.2** (↑ 4.9%) |
| | AVG | 47.0 ± 37.0 | 46.4 ± 35.2 | 40.2 ± 37.6 | 38.9 ± 25.5 | 66.2±33.7 | 81.4 ± 1.4 | 39.8 ± 39.0 | 44.2 ± 34.6 | 41.8±38.8 | 70.6 ± 13.7 | **84.8 ± 2.7** (↑ 4.1%) |
| Citeseer | Clean | 73.6 ± 0.6 | 74.7 ± 1.0 | 74.1 ± 0.9 | 75.2 ± 0.5 | 71.3±1.0 | 74.6 ± 0.5 | 74.9 ± 1.3 | 74.0 ± 1.3 | 73.6±0.9 | 74.7 ± 0.9 | **82.0 ± 1.9** (↑ 9.1%) |
| | nettack | 72.6 ± 0.7 | 72.6 ± 1.8 | 71.5 ± 0.9 | 20.8 ± 8.5 | 70.3±1.1 | 73.2 ± 0.6 | 74.5 ± 1.1 | 71.8 ± 1.5 | 72.8±0.9 | 73.6 ± 1.2 | **78.8 ± 1.6** (↑ 5.7%) |
| | PGD | 52.7 ± 4.5 | 54.5 ± 5.3 | 41.4 ± 4.1 | 17.7 ± 6.2 | 65.4±1.1 | 70.1 ± 1.1 | 48.2 ± 13.9 | 39.1 ± 6.0 | 36.2±1.9 | 60.1 ± 2.5 | **76.9 ± 3.4** (↑ 9.8%) |
| | TDGIA | 23.0 ± 3.8 | 44.7 ± 11.2 | 16.9 ± 2.1 | 15.5 ± 5.3 | 57.1±2.4 | 63.8 ± 7.4 | 28.1 ± 11.1 | 18.2 ± 3.6 | 21.6±1.1 | 57.5 ± 1.7 | **75.9 ± 3.9** (↑ 19.0%) |
| | G-NIA | 15.0 ± 3.6 | 13.6 ± 3.6 | 22.5 ± 4.8 | 18.5 ± 6.6 | 18.2±0.8 | 32.1 ± 6.4 | 54.4 ± 16.8 | 30.2 ± 4.2 | 14.1±1.0 | 68.0 ± 0.9 | **79.4 ± 3.0** (↑ 16.8%) |
| | AVG | 47.4 ± 27.3 | 52.0 ± 24.9 | 45.3 ± 26.7 | 29.5 ± 25.6 | 56.4±22.1 | 62.8 ± 17.7 | 56.0 ± 19.6 | 46.6 ± 25.1 | 43.6±28.1 | 66.8 ± 7.8 | **78.6 ± 2.4** (↑ 17.7%) |
| Reddit | Clean | 84.9 ± 0.6 | 88.5 ± 0.3 | 66.2 ± 3.1 | 83.6 ± 0.4 | 85.7±0.3 | 68.0 ± 1.7 | 50.2 ± 8.3 | 72.7 ± 0.6 | 85.6±0.8 | 86.9 ± 0.4 | **90.8 ± 0.3** (↑ 2.7%) |
| | nettack | 84.8 ± 0.5 | 87.9 ± 0.4 | 68.8 ± 3.1 | 3.6 ± 3.2 | 86.5±0.4 | 67.0 ± 1.7 | 49.5 ± 8.4 | 71.4 ± 0.7 | 84.5±0.8 | 85.5 ± 0.4 | **89.1 ± 0.5** (↑ 1.4%) |
| | PGD | 46.0 ± 1.6 | 30.8 ± 2.5 | 20.0 ± 5.4 | 3.6 ± 1.5 | 81.2±0.8 | 53.1 ± 2.1 | 9.8 ± 3.3 | 19.0 ± 1.0 | 16.5±0.9 | 72.1 ± 0.9 | **81.6 ± 0.9** (↑ 0.4%) |
| | TDGIA | 24.1 ± 1.6 | 32.8 ± 3.8 | 9.0 ± 3.0 | 3.6 ± 1.4 | 48.5±1.5 | 44.3 ± 1.9 | 5.5 ± 1.6 | 8.3 ± 0.6 | 6.3±0.7 | 73.1 ± 0.7 | **81.3 ± 0.6** (↑ 11.1%) |
| | G-NIA | 1.0 ± 0.8 | 2.5 ± 1.1 | 4.0 ± 3.6 | 4.7 ± 2.2 | 8.8±2.6 | 5.0 ± 2.0 | 5.3 ± 3.7 | 3.3 ± 0.7 | 1.9±0.9 | 76.9 ± 1.2 | **84.2 ± 1.1** (↑ 9.5%) |
| | AVG | 48.2 ± 37.1 | 48.5 ± 38.2 | 33.6 ± 31.5 | 19.8 ± 35.6 | 62.1±33.7 | 47.5 ± 25.7 | 24.1 ± 23.6 | 34.9 ± 34.4 | 39.0±42.4 | 78.9 ± 6.9 | **85.4 ± 4.4** (↑ 8.2%) |
| ogbn-products | Clean | 63.9 ± 0.7 | 69.6 ± 0.4 | 49.7 ± 2.4 | 67.6 ± 0.8 | 71.9±0.6 | 64.3 ± 0.4 | 57.7 ± 2.2 | 57.9 ± 0.9 | 72.8±0.4 | 67.6 ± 0.7 | **76.1 ± 0.4** (↑ 4.5%) |
| | nettack | 63.3 ± 0.6 | 62.1 ± 2.1 | 50.1 ± 3.1 | 14.9 ± 1.4 | 72.8±0.7 | 62.1 ± 0.8 | 56.1 ± 2.4 | 52.6 ± 0.9 | 71.3±0.5 | 65.8 ± 0.5 | **74.4 ± 0.6** (↑ 2.3%) |
| | PGD | 32.2 ± 0.9 | 25.0 ± 0.9 | 17.5 ± 1.0 | 14.3 ± 1.8 | 57.7±2.7 | 34.8 ± 0.9 | 17.8 ± 1.5 | 21.2 ± 0.5 | 19.8±0.7 | 54.0 ± 0.6 | **67.9 ± 0.6** (↑ 17.5%) |
| | TDGIA | 23.1 ± 1.0 | 16.9 ± 1.1 | 11.0 ± 0.5 | 16.5 ± 1.9 | 54.6±2.4 | 28.0 ± 0.9 | 16.5 ± 2.2 | 16.9 ± 0.8 | 11.7±0.5 | 49.5 ± 1.0 | **64.9 ± 0.9** (↑ 18.8%) |
| | G-NIA | 2.7 ± 1.0 | 3.6 ± 2.0 | 2.2 ± 0.8 | 9.9 ± 5.2 | 4.4±1.7 | 7.1 ± 2.6 | 8.6 ± 4.5 | 3.3 ± 0.6 | 0.8±0.3 | 54.2 ± 0.8 | **65.6 ± 1.1** (↑ 21.0%) |
| | AVG | 37.1 ± 26.5 | 35.5 ± 28.9 | 26.1 ± 22.4 | 24.7 ± 24.1 | 52.3±28.0 | 39.3 ± 24.1 | 31.3 ± 23.6 | 30.4 ± 23.7 | 35.3±34.2 | 58.2 ± 8.0 | **69.8 ± 5.2** (↑ 19.8%) |
| ogbn-arxiv | Clean | 65.3 ± 0.3 | 65.2 ± 0.1 | - | - | 53.0±0.1 | 60.2 ± 1.0 | - | 58.0 ± 0.1 | 61.1±0.2 | 61.0 ± 0.7 | **66.7 ± 0.4** (↑ 2.1%) |
| | PGD | 41.1 ± 1.0 | 22.6 ± 1.9 | - | - | 52.3±0.1 | 37.8 ± 2.0 | - | 29.9 ± 0.6 | 19.1±0.4 | 24.2 ± 2.8 | **52.9 ± 1.0** (↑ 1.2%) |
| | TDGIA | 33.1 ± 1.6 | 9.7 ± 1.8 | - | - | 51.6±0.1 | 27.5 ± 2.1 | - | 20.5 ± 0.9 | 18.6±0.8 | 29.3 ± 2.3 | **53.2 ± 0.8** (↑ 3.0%) |
| | G-NIA | 4.6 ± 0.4 | 2.5 ± 0.3 | - | - | 35.3±0.1 | 5.6 ± 0.8 | - | 14.6 ± 0.1 | 2.5±0.1 | 11.5 ± 1.1 | **40.5 ± 1.6** (↑ 14.5%) |
| | AVG | 36.0 ± 25.0 | 25.0 ± 28.1 | - | - | 48.1±8.5 | 32.8 ± 22.7 | - | 30.7 ± 19.2 | 27.3±22.2 | 31.5 ± 21.0 | **53.3 ± 10.7** (↑ 10.9%) |

The proof of Proposition 2 is available in Appendix B.2. Note that condition (1) aligns with minimizing the losses $\mathcal{L}_\mathcal{P}$, $\mathcal{L}_\mathcal{I}$, and $\mathcal{L}_\mathcal{E}$ in Eq.11. In condition (2), the first term corresponds to minimizing $\mathcal{L}_\mathcal{D}$ in Eq.11, while the second term implies the diversity of adversarial examples, common in graph. Proposition 2 serves as a theoretical validation for the effectiveness of IDEA.

# 4 EXPERIMENTS

**Datasets.** To evaluate the adaptability of IDEA across various datasets, we conduct node classification experiments on 5 diverse network benchmarks. These include three citation networks: Cora (Bojchevski & Günnemann, 2019), Citeseer (Bojchevski & Günnemann, 2019), and obgn-arxiv (Hu et al., 2020), a social network Reddit (Hamilton et al., 2017; Zeng et al., 2020), as well as a product co-purchasing network ogbn-products (Hu et al., 2020). The statistics of datasets are in Appendix D.1.

**Attack methods and defense baselines.** To demonstrate the effectiveness of our IDEA, we compare IDEA with the state-of-the-art defense methods. Specifically, traditional GNNs including GCN (Kipf & Welling, 2017) and GAT (Veličković et al., 2018); graph purification including ProGNN (Jin et al., 2020b), STABLE (Li et al., 2022c), and GARNET (Deng et al., 2022); robust aggregation including RGCN (Zhu et al., 2019), SimPGCN (Jin et al., 2021), Elastic (Liu et al., 2021), and Soft-Median (Geisler et al., 2021), as well as adversarial training FLAG (Kong et al., 2022). The details are described in Appendix D.2. We evaluate the robustness of IDEA using five attacks, including a representative poisoning attack MetaAttack (Zügner & Günnemann, 2019b) and four evasion attacks, i.e., nettack (Zügner et al., 2018), PGD (Madry et al., 2018), TDGIA (Zou et al., 2021), G-NIA (Tao et al., 2021b). The details of attacks are provided in Appendix D.3.

**Implementation.** For each dataset, we randomly split nodes as 1:1:8 for training, validation and test, following (Jin et al., 2020b; 2021; Liu et al., 2021; Li et al., 2022c). For each experiment, we report the average performance and the standard deviation of 10 runs. We employ the widely-used DeepRobust (Li et al., 2021) library for the attack and defense methods. We tune their hyper-parameters according validation set. Note that MetaAttack is untargeted attack, performance is reported on the test set with perturbation rates from 0% to 20%, following (Liu et al., 2021; Li et al., 2022c). The evasion attacks are targeted attacks, and we randomly sample 20% of all nodes from the test set as targets. Nettack perturbs 20% edges, while node injection attacks (PGD, TDGIA, and G-NIA) inject 20% nodes and edges. We focused on gray-box attack scenarios following (Zügner et al., 2018; Zügner & Günnemann, 2019b; Li et al., 2022c; Jin et al., 2021; 2020b) and the attacker is only aware of the input and output, which is practical. For IDEA, our backbone model is GCN, which is used for the encoder $h$. We tune the hyper-parameters from the following range: the coefficient $\alpha$ over $\{10, 25, 100, 150\}$, the number of domains over $\{2, 5, 10, 20\}$. The details are in Appendix D.4.

Table 2: Accuracy(%) of test set under poisoning attack (MetaAttack).

| Dataset | Pb. rate | GCN | GAT | ProGNN | STABLE | GARNET | RGCN | SimpGCN | Elastic | Soft-Median | FLAG | IDEA |
|---|---|---|---|---|---|---|---|---|---|---|---|---|
| Cora | 0% | 83.6±0.5 | 83.5±0.5 | 83.0±0.2 | 85.6±0.6 | 80.1±0.5 | 82.6±0.3 | 81.9±1.0 | 85.8±0.4 | 84.0±0.5 | 83.4±0.3 | **87.1**±0.7 (↑1.5%) |
| | 5% | 77.8±0.6 | 80.3±0.5 | 82.3±0.5 | 81.4±0.5 | 77.1±0.8 | 77.5±0.5 | 77.6±0.7 | 82.2±0.9 | 79.9±0.8 | 80.9±0.3 | **85.5**±0.6 (↑3.9%) |
| | 10% | 74.9±0.7 | 78.5±0.6 | 79.0±0.6 | 80.5±0.6 | 75.3±0.8 | 73.7±1.2 | 75.7±1.1 | 78.8±1.7 | 73.4±2.3 | 78.8±0.9 | **84.8**±0.6 (↑5.3%) |
| | 15% | 67.8±1.2 | 73.6±0.8 | 76.4±1.3 | 78.6±0.4 | 72.3±0.7 | 70.2±0.6 | 72.7±2.8 | 77.2±1.6 | 70.5±1.1 | 75.0±0.7 | **84.2**±0.6 (↑7.1%) |
| | 20% | 61.6±1.1 | 66.6±0.8 | 73.3±1.6 | 77.8±1.1 | 69.8±0.7 | 62.7±0.7 | 70.3±4.6 | 70.5±1.3 | 60.5±0.4 | 70.2±1.1 | **83.2**±0.6 (↑6.9%) |
| Citeseer | 0% | 73.3±0.3 | 74.4±0.8 | 73.3±0.7 | 75.8±0.4 | 70.4±0.7 | 74.4±0.3 | 74.4±0.7 | 73.8±0.6 | 71.3±0.8 | 72.8±0.8 | **80.3**±1.0 (↑5.9%) |
| | 5% | 70.2±0.8 | 72.3±0.5 | 72.9±0.6 | 74.1±0.6 | 69.2±0.9 | 71.7±0.3 | 73.3±1.0 | 72.9±0.5 | 69.6±2.2 | 71.1±0.6 | **79.1**±0.9 (↑6.7%) |
| | 10% | 68.0±1.4 | 70.3±0.7 | 72.5±0.8 | 73.5±0.4 | 68.5±1.0 | 69.3±0.4 | 72.0±1.0 | 72.6±0.4 | 67.9±1.9 | 69.2±0.6 | **78.6**±1.1 (↑6.9%) |
| | 15% | 65.2±0.9 | 67.7±1.0 | 72.0±1.1 | 73.2±0.5 | 65.0±1.2 | 66.0±0.2 | 70.8±1.3 | 71.9±0.7 | 66.0±2.9 | 66.5±0.8 | **77.6**±0.6 (↑6.0%) |
| | 20% | 60.1±1.4 | 64.3±1.0 | 70.0±2.3 | 72.8±0.5 | 62.9±1.9 | 61.2±0.5 | 70.0±1.7 | 64.7±0.8 | 56.1±1.3 | 64.1±0.8 | **77.8**±0.9 (↑6.9%) |
| Reddit | 0% | 84.5±0.5 | 88.0±0.3 | 73.4±2.8 | 86.6±0.2 | 85.2±0.2 | 78.2±0.6 | 51.4±7.6 | 83.8±0.3 | 88.8±0.5 | 84.6±0.2 | **91.2**±0.3 (↑2.7%) |
| | 5% | 81.0±0.8 | 86.1±0.6 | 73.9±1.2 | 81.5±0.4 | 78.3±0.3 | 73.9±1.7 | 34.8±9.9 | 80.6±0.5 | 82.6±0.9 | 83.6±0.4 | **90.0**±0.4 (↑4.5%) |
| | 10% | 72.1±0.6 | 78.8±0.8 | 63.2±1.2 | 75.9±0.5 | 63.4±1.3 | 57.8±1.3 | 27.3±7.8 | 70.4±1.0 | 66.4±1.4 | 72.2±0.8 | **89.0**±0.4 (↑12.9%) |
| | 15% | 70.1±1.7 | 76.0±1.6 | 59.9±1.4 | 73.8±0.4 | 62.2±0.6 | 53.3±1.4 | 25.0±6.4 | 68.7±0.6 | 60.5±2.4 | 66.3±1.2 | **88.4**±0.4 (↑16.3%) |
| | 20% | 67.9±1.4 | 72.7±1.9 | 56.7±0.9 | 71.7±0.5 | 60.7±0.8 | 51.5±2.8 | 19.0±4.5 | 67.4±0.6 | 56.6±1.5 | 63.7±0.5 | **88.1**±0.3 (↑21.2%) |
| ogbn-products | 0% | 63.0±0.7 | 68.6±0.4 | 64.3±2 | 70.5±0.5 | 71.3±0.5 | 63.0±0.5 | 57.1±2.1 | 72.7±0.2 | 74.3±0.3 | 66.3±0.4 | **75.2**±0.3 (↑1.2%) |
| | 5% | 49.6±0.9 | 64.4±0.3 | 50.0±2.4 | 58.7±0.7 | 61.5±0.6 | 40.0±1.1 | 31.9±10.3 | 61.0±0.7 | 66.5±0.4 | 57.3±0.7 | **73.7**±0.5 (↑10.8%) |
| | 10% | 39.4±1.1 | 54.4±0.7 | 43.4±1.9 | 50.5±0.5 | 53.6±0.9 | 33.4±0.9 | 26.7±8.1 | 52.3±0.7 | 56.3±0.7 | 46.7±0.7 | **72.9**±0.3 (↑29.5%) |
| | 15% | 34.4±1.1 | 46.7±0.7 | 38.4±1.8 | 44.6±0.8 | 47.7±0.4 | 29.9±1.0 | 20.3±6.6 | 48.6±0.4 | 47.7±1.0 | 42.7±0.5 | **71.9**±0.7 (↑47.9%) |
| | 20% | 31.2±1.0 | 41.9±1.2 | 34.0±2.5 | 40.3±0.7 | 42.3±0.4 | 27.8±0.8 | 16.0±2.1 | 46.3±0.4 | 42.8±1.3 | 39.3±0.6 | **71.0**±0.8 (↑53.3%) |

## 4.1 ROBUSTNESS AGAINST EVASION ATTACKS

We conduct experiments under four evasion attacks (nettack, PGD, TDGIA, and G-NIA), and show the accuracy of target nodes in Table 1. We also report the average accuracy of clean and attacked graphs, along with standard deviation of accuracy across these graphs, denoted as AVG. Note that we exclude nettack from ogbn-arxiv evaluation due to its lack of scalability. GCN and GAT exhibit high accuracy on clean graphs, however, their accuracy significantly declines under PGD, TDGIA, and G-NIA. Defense methods suffer from severe performance degradation under various attacks, and some (such as ProGNN (81.9%) and RGCN (83.6%)) even experience a decline on Clean. For graph purification methods, ProGNN and STABLE perform poorly under most attacks, maybe because they require retraining to achieve defensive effects, rendering them unsuitable for evasion attacks. GARNET shows effectiveness against PGD and TDGIA, but still struggles to defend against G-NIA. RGCN, SimPGCN, Elastic, and Soft-Median perform well against nettack; however, they suffer from performance degradation on clean graphs, which is undesirable. Adversarial training FLAG outperforms other baselines but exhibits unsatisfactory defense on Cora, Citeseer, and ogbn-arxiv.

Our proposed method, IDEA, achieves the best performance on Clean and across all attacks, significantly outperforming all baselines on all datasets. On Clean, IDEA exhibits the best performance primarily due to its ability to learn causal features that have strong label predictability. Furthermore, IDEA's performance remains good consistency under both clean graphs and various attacks, evidenced by the low standard deviation in AVG. This emphasizes its invariant predictability across all attacks. For instance, on Citeseer, IDEA's standard deviation across graphs is a only 2.4, while the runner-up, FLAG, reaches 7.8. These results demonstrate that IDEA possesses both strong predictability (high accuracy on Clean) and invariant predictability (sustained accuracy across attacks).

## 4.2 ROBUSTNESS AGAINST POISONING ATTACKS

We evaluate IDEA's robustness under poisoning attacks, employing the widely-adopted MetaAttack (Zügner & Günnemann, 2019b) and varying the perturbation rate (the rate of changing edges) from 0 to 20% following (Liu et al., 2021; Li et al., 2022c). We exclude ogbn-arxiv since MetaAttack cannot handle large graphs. Table 2 shows that all methods' accuracy decreases as perturbation rate increases. Among baselines, graph purification methods demonstrate better defense performance, with STABLE outperforming others on Cora and Citeseer. RGCN, SimPGCN, and Soft-Median resist attacks only at low perturbation rates. The adversarial training FLAG brings less improvement than it does under evasion attacks. Maybe due to its training on evasion attacks, leading to poor generalization for poisoning attacks.

For ours, IDEA also achieves the state-of-the-art performance under all perturbation rates on all datasets, significantly outperforming all baselines. When the attack strength becomes larger, our IDEA still maintains good performance, demonstrating that IDEA has the invariant prediction ability across perturbations.

## 4.3 ABLATION STUDY

We analyze the influence of each part of IDEA through experiments on invariance goals and domain construction. We implement four variants of IDEA, including IDEA removing the node-based

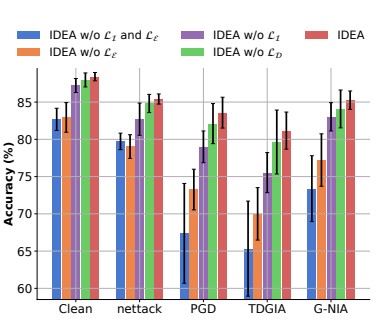

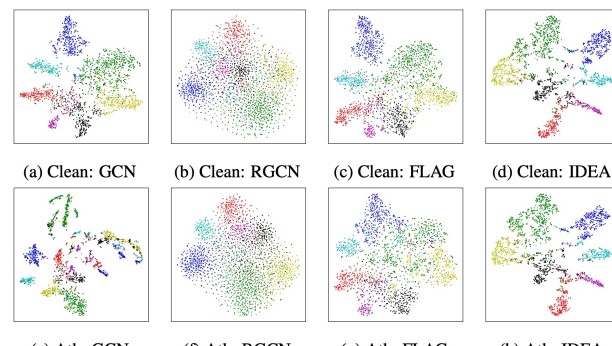

Figure 4: Ablation Study.

Figure 5: Visualizing learned features: clean and attacked graphs.

invariance goal (IDEA w/o $\mathcal{L}_{\mathcal{I}}$), IDEA removing the structure-based invariance goal $\mathcal{L}_{\mathcal{E}}$ (IDEA w/o $\mathcal{L}_{\mathcal{E}}$), IDEA removing both invariance goals (IDEA w/o $\mathcal{L}_{\mathcal{I}}$ and $\mathcal{L}_{\mathcal{E}}$), and IDEA removing domain partition (IDEA w/o $\mathcal{L}_{\mathcal{D}}$). IDEA w/o $\mathcal{L}_{\mathcal{I}}$ and $\mathcal{L}_{\mathcal{E}}$ only optimizes the predictive loss, which can be regarded as an adversarial training version of IDEA, which can verify the benefit brought by learning causal features. We take the results on clean graph and evasion attacks on Cora as an illustration.

As shown in Figure 4, all variants exhibit a decline compared to IDEA (red), highlighting the significance of both invariance goals and domain construction. Specifically, IDEA w/o $\mathcal{L}_{\mathcal{I}}$ and $\mathcal{L}_{\mathcal{E}}$ (blue) suffers the largest drop, highlighting our objectives' benefits since IDEA is much more robust than simple adversarial training using same adversarial examples. The performance decline of IDEA w/o $\mathcal{L}_{\mathcal{E}}$ (orange) illustrates the significant advantages of the structural-based invariance goal, especially on clean graph, highlighting the benefits of modeling the interactions between samples. IDEA w/o $\mathcal{L}_{\mathcal{D}}$ (green) displays a large standard deviation, with the error bar much larger than that of IDEA, emphasizing the stability achieved through the diverse attack domains.

Detailed hyper-parameter analysis regarding coefficient $\alpha$ and the number of attack domains can be found in Appendix D.5. The performance under adaptive attacks are shown in Appendix D.6.

## 4.4 VISUALIZATION

We further visualize the learned features with t-SNE technique (Van der Maaten & Hinton, 2008) to show whether IDEA learns the features that have strong and invariant predictability. Figure 5 illustrates the feature learned by GCN, RGCN, FLAG, and IDEA on clean graph and under the strongest G-NIA attack on Cora. As shown in Figure 5, existing methods either learn discriminative features on Clean but destroyed under attack (GCN and FLAG), or learn features are mixed (RGCN).

For our IDEA, in Figure 5(d,h), the features learned by IDEA can be distinguished by labels. Specifically, IDEA's learned features are similar for nodes with the same label and distinct for different labels, emphasizing features' **strong predictability** for labels. Furthermore, the features in Figure 5(d) on the clean graph and those in Figure 5(h) on the attacked graph exhibit nearly the same distributions. This observation demonstrates that the relationship between features and labels can remain invariant across attacks, thus exhibiting **invariant predictability**. These results show that IDEA learned causal invariant features with both strong and invariant predictability for labels.

## 5 CONCLUSION AND FUTURE WORK

In this paper, we creatively introduce a causal defense perspective by learning causal features that have strong and invariant predictability across attacks. Then, we propose IDEA and design two invariance objectives to learn causal features. Extensive experiments demonstrate that IDEA significantly outperforms all the baselines under both evasion attacks and poisoning attacks on five benchmark datasets, emphasizing that IDEA possesses both strong and invariant predictability across attacks. We believe causal defense approach is a promising new direction, and there are many interesting and valuable research problems in the future. For example, studying domain partitioning is more suitable for adversarial attack and defense scenarios; or exploring more ways to generate adversarial examples, and so on.

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

# A  RELATED WORKS

In this section, we present the related works on defense methods against graph adversarial attacks and invariant learning methods.

## A.1  DEFENSE AGAINST GRAPH ADVERSARIAL ATTACK

Despite the success of graph neural networks (GNNs), they are shown to be vulnerable to adversarial attacks, (Zügner et al., 2018; Sun et al., 2018; Chen et al., 2020), i.e., imperceptible perturbations on graph data can dramatically degrade the performance of GNNs (Tao et al., 2021b; Zou et al., 2021; Sun et al., 2020; Wang et al., 2020a; Tao et al., 2023a), blocking the deployment of GNNs to real world applications (Jin et al., 2020a). Various defense mechanisms (Li et al., 2023; Gosch et al., 2023; Tao et al., 2023b; 2021a) have been proposed to counter these graph adversarial attacks, which can broadly be classified into adversarial training, graph purification, and robust aggregation strategies (Sun et al., 2018; Li et al., 2022c; Jin et al., 2020a).

Adversarial training methods, such as FLAG (Kong et al., 2022) and others (Dai et al., 2019; Feng et al., 2019; Li et al., 2022b), typically employ a min-max optimization approach. This involves iteratively generating adversarial examples that maximize the loss and updating GNN parameters to minimize the loss on these examples. However, adversarial training may be not robust under unseen attacks (Bojchevski & Günnemann, 2019). Robust training methods (Zügner & Günnemann, 2019a; Bojchevski & Günnemann, 2019; Zügner & Günnemann, 2020; Bojchevski et al., 2020; Schuchardt et al., 2021) incorporate worst-case adversarial examples to enhance certifiable robustness (Bojchevski & Günnemann, 2019; Zügner & Günnemann, 2019a). These methods can be considered an improved version of traditional adversarial training. However, due to limited searching space, robust training still faces similar challenges as adversarial training.

Graph purification methods (Wu et al., 2019; Jin et al., 2020b; Entezari et al., 2020) aim to purify adversarial perturbations by modifying graph structure. Jaccard (Wu et al., 2019) prunes edges that connect two dissimilar nodes, while ProGNN (Jin et al., 2020b) concurrently learns the graph structure and GNN parameters through optimization of feature smoothness, low-rank and sparsity. The recent method STABLE (Li et al., 2022c) acquires reliable representations of graph structure via unsupervised learning. GARNET (Deng et al., 2022) first leverages weighted spectral embedding to construct a base graph, then refines the base graph by pruning additional uncritical edges based on probabilistic graphical model, to boost the adversarial robustness of GNN models.

Robust aggregation methods (Zhu et al., 2019; Liu et al., 2021; Jin et al., 2021; Lei et al., 2022; Zhang & Zitnik, 2020) redesign model structures to establish robust GNNs. RGCN (Zhu et al., 2019) uses Gaussian noise to mitigate adversarial perturbations. SimPGCN (Jin et al., 2021) resents a feature similarity preserving aggregation that balances the structure and feature information. Elastic (Liu et al., 2021) improves the local smoothness adaptivity and derives the elastic message passing. Geisler et al. (Geisler et al., 2021) design a robust aggregation function, Soft Median to achieve an effective defense at all scales. However, both kinds of methods rely on specific heuristic priors such as local smoothness (Wu et al., 2019; Veličković et al., 2018; Jin et al., 2020b; Zhang & Zitnik, 2020; Li et al., 2022c; Jin et al., 2021) or low rank (Jin et al., 2020b; Entezari et al., 2020), that may be ineffective against some attacks (Chen et al., 2022b), leading to method failure. What's worse, modifying graph structure (Jin et al., 2020b; Zhang & Zitnik, 2020) or adding noise (Zhu et al., 2019) with this heuristic may even cause performance degradation on clean graphs.

Different from the above studies, in this paper, we creatively propose an invariant causal defense perspective, providing a new perspective to address this issue. Our method aims to learn causal features that possess strong predictability for labels and invariant predictability across attacks, to achieve graph adversarial robustness.

## A.2  INVARIANT LEARNING METHODS

Invariant learning methods (Arjovsky et al., 2019; Krueger et al., 2021; Li et al., 2022a) have fueled a surge of research interests (Shen et al., 2021; Creager et al., 2021; Yong et al., 2022; Chen et al., 2022a;c; Wu et al., 2022). These work typically assume that data are collected through different domains or environments (Arjovsky et al., 2019), and the causal relationships within the data remain

unchanged across different domains, denoting invariant causality (Shen et al., 2021). Generally, invariant learning methods aim to learn the causal mechanism or causal feature that is invariant across different domains or environments, allowing the causal feature to generalize across all domains, which can be used to solve the out-of-distribution generalization problem (Shen et al., 2021; Arjovsky et al., 2019; Rosenfeld et al., 2021).

However, such methods cannot be directly applied to solve graph adversarial robustness due to the complex nature of graph data and the scarcity of diverse domains. Two main challenges arise: i) On graph data, there are interconnections (edges) between nodes, so nodes are no longer independent of each other, making samples not independent and identically distributed (non-IID) (Wu et al., 2022; Chen et al., 2022c). We model the generation of graph adversarial attack via an interaction causal model and propose corresponding invariance goals considering both node itself and the interconnection between nodes. ii) In adversarial learning, constructing sufficiently diverse domains or environments is challenging due to a lack of varied domains. We propose to learn sufficient and diverse domains by limiting the co-linearity between domains.

### A.3   CAUSAL METHODS FOR ADVERSARIAL ROBUSTNESS

A few recent works attempt to achieve adversarial robustness with causal methods on computer vision (Ren et al., 2022; Zhang et al., 2022b). These methods, such as DICE (Ren et al., 2022), ADA (Zhang et al., 2022b), mainly use causal intervention to achieve the robustness. The difference between them and our work lies in two aspects: (1) Existing causal methods for robustness are developed for the image area. However, the non-IID nature of graph data brings challenges to these methods in achieving graph adversarial robustness. Our work proposes the structural-level invariance goal for the non-IID graph data. (2) These methods adopt causal intervention. For example, DICE uses hard intervention (Ren et al., 2022), and ADA (Zhang et al., 2022b) uses "soft" intervention. However, the intervention is difficult to achieve (Pearl, 2009). Our work constructs diverse domains and learns causal features by optimizing both node-based and structural-based invariance goals.

### A.4   PURIFICATION METHODS IN COMPUTER VISION

There are also some purification works in computer vision for defending against attacks. Shi et al. propose Self-supervised Online Adversarial Purification (SOAP), leveraging self-supervised loss to purify adversarial examples at test-time (Shi et al., 2021). Zhou et al. propose to remove adversarial noise by implementing a self-supervised adversarial training mechanism in a class activation feature space (Zhou et al., 2021). Naseer et al. propose a self-supervised adversarial training mechanism in the input space (Naseer et al., 2020). Liao et al. propose high-level representation guided denoiser (HGD), using a loss function defined as the difference between the target model's outputs activated by the clean image and denoised image (Liao et al., 2018).

Most purification methods in computer vision leverage image data priors. For example, SOAP (Shi et al., 2021) incorporates self-supervised tasks such as image rotation that are unique to the domain of computer vision, while NRP (Naseer et al., 2020) depends on a pixel loss function, i.e., $\mathcal{L}_{img}$, to encourage image smoothness. These domain-specific dependencies pose significant challenges when considering the direct transposition of these methods to graph data, which inherently lacks such image-based priors.

In contrast, graph purification methods (Jin et al., 2020b; Li et al., 2022c) are specifically designed to exploit the unique properties of graph data, making them appropriate for addressing graph-specific issues. However, graph purification defenses rely on predefined heuristics, while these may be ineffective for som attacks causing the methods to fail. Therefore, there is a pressing need to develop a defense strategy that is robust and effective against various attacks.

# B PROOFS

## B.1 PROOF FOR PROPOSITION 1

*Proof.* The difference between $\hat{I}(Y, D|Z)$ and $I(Y, D|Z)$ could be written as

$$
\begin{aligned}
&\hat{I}(Y, D|Z) - I(Y, D|Z) \\
&= \mathbb{E}_{p(z)}\left[\mathbb{E}_{p(y,d|Z)}\left[\left[\log q_d(y \mid z, d) - \log q(y|z)\right] - \left[\log p(y \mid z, d) - \log p(y \mid z)\right]\right]\right] \\
&= \mathbb{E}_{p(z)}\left[\mathbb{E}_{p(y,d|z)}\left[\log \frac{p(y \mid z)}{q(y \mid z)} - \log \frac{p(y \mid z, d)}{q_d(y \mid z, d)}\right]\right] \\
&= \mathbb{E}_{p(z)}\left[\mathbb{E}_{p(y|z)}\left[\log \frac{p(y \mid z)}{q(y \mid z)}\right] - \mathbb{E}_{p(d|z)}\mathbb{E}_{p(y|z,d)}\left[\log \frac{p(y \mid z, d)}{q_d(y \mid z, d)}\right]\right] \\
&= \mathbb{E}_{p(z)}KL\left[p(y \mid z)\|q(y \mid z)\right] - \mathbb{E}_{p(z,d)}KL\left[p(y \mid z, d)\|q_d(y \mid z, d)\right]
\end{aligned}
\tag{12}
$$

Next, similar to the theoretical analysis in CLUB (Cheng et al., 2020), we can prove that $\hat{I}$ is either a upper bound of $I$ or a esitimator of $I$ whose absolute error is bounded by the approximation performance $\mathbb{E}_{p(z,d)}KL[p(y \mid z, d)\|q_d(y \mid z, d)]$. That is to say, if $\mathbb{E}_{p(z,d)}KL[p(y \mid z, d)\|q_d(y \mid z, d)]$ is small enough, $I(Y, D|Z)$ is bounded by $\hat{I}(Y, D|Z)$. Therefore, $\hat{I}(Y, D|Z)$ is minimized if $\mathbb{E}_{p(z,d)}KL[p(y \mid z, d)\|q_d(y \mid z, d)]$ and $\hat{I}(Y, D|Z)$ are both minimized. $\qquad\square$

## B.2 PROOF FOR PROPOSITION 2

*Proof.* The proof includes three steps:

**Step 1:** We prove that if $\Phi$ and $\omega$ satisfies the condition (1), i.e., $I\left(\Phi(\hat{G}), Y\right) - \left[I\left(Y, D \mid \Phi(\hat{G})\right) + I\left(Y, D \mid \Phi(\hat{G})_{\mathcal{N}}\right)\right]$ (denoted as $\kappa$) is maximized, then $\Theta_\phi\mathbb{E}_{\rho(\hat{G})^D}\left[\rho(\hat{G}^D)\rho(\hat{G}^D)^\top\right]\Theta_\phi^\top\Theta_\omega = \Theta_\phi\mathbb{E}_{\rho(\hat{G}^D),Y^D}\left[\rho(\hat{G}^D)Y^D\right]$ where $\hat{G}^D = \left\{\rho(\hat{G})_i|i \in \mathcal{V}^D\right\}$, for all $D \in \mathcal{D}_{\text{tr}}$. Next, we begin our proof. Suppose that $\Phi$ has infinite capacity for representation, with $\Phi = \arg\max \kappa$ and $\Phi$ containing $\phi$ and $\rho$, we have $Y^D = \rho(\hat{G}^D)^\top\Theta_\phi^\top\Theta_\omega + \epsilon_\Phi$, where $\rho(\hat{G}^D)^\top\Theta_\phi^\top\Theta_\omega$ represents the output of $\rho(\hat{G}^D)$ after passing through learner $\phi$ and classifier $\omega$ (i.e., the output of $\hat{G}^D$ after passing through $\Phi$ and $\omega$). The error term $\epsilon_\Phi$ satisfies $\mathbb{E}_{Y^D}[\epsilon_\Phi] = 0$. We have:

$$
\begin{aligned}
Y^D &= \rho(\hat{G}^D)^\top\Theta_\phi^\top\Theta_\omega + \epsilon_\Phi \\
\mathbb{E}_{Y^D}\left[Y^D\right] &= \mathbb{E}_{Y^D}\left[\rho(\hat{G}^D)^\top\Theta_\phi^\top\Theta_\omega + \epsilon_\Phi\right] = \mathbb{E}_{Y^D}\left[\rho(\hat{G}^D)^\top\Theta_\phi^\top\Theta_\omega\right] = \rho(\hat{G}^D)^\top\Theta_\phi^\top\Theta_\omega \\
\mathbb{E}_{\rho(\hat{G}^D),Y^D}\left[\rho(\hat{G}^D)Y^D\right] &= \mathbb{E}_{\rho(\hat{G}^D)}\left[\rho(\hat{G}^D)\rho(\hat{G}^D)^\top\right]\Theta_\phi^\top\Theta_\omega \\
\Theta_\phi\mathbb{E}_{\rho(\hat{G}^D),Y^D}\left[\rho(\hat{G}^D)Y^D\right] &= \Theta_\phi\mathbb{E}_{\rho(\hat{G}^D)}\left[\rho(\hat{G}^D)\rho(\hat{G}^D)^\top\right]\Theta_\phi^\top\Theta_\omega.
\end{aligned}
\tag{13}
$$

The validity of line 2 in Eq. 13 stems from $\mathbb{E}_{Y^D}[\epsilon_\Phi] = 0$, and the fact that $Y^D$ is independent with $\rho(\hat{G}^D)^\top\Theta_\phi^\top\Theta_\omega$. Consequently, we have $\Theta_\phi\mathbb{E}_{\rho(\hat{G}^D)}\left[\rho(\hat{G}^D)\rho(\hat{G}^D)^\top\right]\Theta_\phi^\top\Theta_\omega = \Theta_\phi\mathbb{E}_{\rho(\hat{G}^D),Y^D}\left[\rho(\hat{G}^D)Y^D\right]$.

**Step 2:** We prove that if $\Theta_\phi^\top\Theta_\omega$ satisfies the condition (2), i.e., $\left\{\mathbb{E}_{\rho(\hat{G}^D)}\left[\rho(\hat{G}^D)\rho(\hat{G})^{D^\top}\right]\left(\Theta_\phi^\top\Theta_\omega - \Theta_{\check{\psi}}^\top\Theta_\gamma\right)\right\}_{D \in \mathcal{D}_{\text{tr}}}$ is linearly independent, and $\dim\left(\text{span}\left(\left\{\mathbb{E}_{\hat{G}_i}\left[\rho(\hat{G})_i\rho(\hat{G})_i^\top\right]\left(\Theta_\phi^\top\Theta_\omega - \Theta_{\check{\psi}}^\top\Theta_\gamma\right)\right\}_{i \in \mathcal{V}}\right)\right) > \dim(\phi) - r$, then $\dim\left(\text{span}\left(\left\{\mathbb{E}_{\rho(\hat{G}^D)}[\rho(\hat{G}^D)\rho(\hat{G}^D)^\top]\left(\Theta_\phi^\top\Theta_\omega - \Theta_{\check{\psi}}^\top\Theta_\gamma\right) - \mathbb{E}_{\rho(\hat{G}^D),\epsilon^D}[\rho(\hat{G}^D)\epsilon^D]\right\}_{D \in \mathcal{D}_{\text{tr}}}\right)\right) > \dim(\phi) - r$.

We examine the two component individually. Suppose that

$$\dim\left(\operatorname{span}\left\{\mathbb{E}_{\rho(\hat{G}^D)}[\rho(\hat{G}^D)\rho(\hat{G}^D)^\top]\left(\Theta_\phi^\top\Theta_\omega - \Theta_{\tilde\psi}^\top\Theta_\gamma\right)\right\}_{D\in\mathcal{D}_{\text{tr}}}\right) = k \tag{14}$$

Since the set $\left\{\mathbb{E}_{\rho(\hat{G}^D)}[\rho(\hat{G}^D)\rho(\hat{G}^D)^\top]\left(\Theta_\phi^\top\Theta_\omega - \Theta_{\tilde\psi}^\top\Theta_\gamma\right)\right\}_{D\in\mathcal{D}_{\text{tr}}}$ is linearly independent, and $\dim\left(\operatorname{span}\left(\left\{\mathbb{E}_{\hat{G}_i}\left[\rho(\hat{G})_i\rho(\hat{G})_i^\top\right]\left(\Theta_\phi^\top\Theta_\omega - \Theta_{\tilde\psi}^\top\Theta_\gamma\right)\right\}_{i\in\mathcal{V}}\right)\right) > \dim(\phi) - r$, we have $k > \dim(\phi) - r$.

Next, we consider $\mathbb{E}_{\rho(\hat{G}^D),\epsilon^D}[\rho(\hat{G}^D)\epsilon^D]$. Since $\operatorname{rank}(A) \geq \operatorname{rank}(AB)$, and both $\epsilon^D$ and $\left(\Theta_\phi^\top\Theta_\omega - \Theta_{\tilde\psi}^\top\Theta_\gamma\right)$ are scalar values that do not affect the dimension, we have

$$\begin{aligned}
&\dim\left(\operatorname{span}\left(\left[\mathbb{E}_{\rho(\hat{G}^D),\epsilon^D}[\rho(\hat{G}^D)\epsilon^D]\right]\right)\right)\\
&\geq \dim\left(\operatorname{span}\left\{\mathbb{E}_{\rho(\hat{G}^D)}[\rho(\hat{G}^D)\rho(\hat{G}^D)^\top]\left(\Theta_\phi^\top\Theta_\omega - \Theta_{\tilde\psi}^\top\Theta_\gamma\right)\right\}_{D\in\mathcal{D}_{\text{tr}}}\right) = k
\end{aligned} \tag{15}$$

Taking the dimensions of both components into account, we arrive at

$$\begin{aligned}
&\dim\left(\operatorname{span}\left(\{\mathbb{E}_{\rho(\hat{G}^D)}[\rho(\hat{G}^D)\rho(\hat{G}^D)^\top]\left(\Theta_\phi^\top\Theta_\omega - \Theta_{\tilde\psi}^\top\Theta_\gamma\right) - \mathbb{E}_{\rho(\hat{G}^D),\epsilon^D}[\rho(\hat{G}^D)\epsilon^D]\}_{D\in\mathcal{D}_{\text{tr}}}\right)\right)\\
&\geq k > \dim(\phi) - r.
\end{aligned} \tag{16}$$

**Step 3:** We prove that if $\Theta_\phi^\top\Theta_\omega$ satisfies: $\Theta_\phi\mathbb{E}_{\rho(\hat{G}^D)}[\rho(\hat{G}^D)\rho(\hat{G}^D)^\top]\Theta_\phi^\top\Theta_\omega = \Theta_\phi\mathbb{E}_{\rho(\hat{G}^D),Y^D}[\rho(\hat{G}^D)Y^D]$, for all $D \in \mathcal{D}_{tr}$ and $\dim(\operatorname{span}(\{\mathbb{E}_{\rho(\hat{G}^D)}[\rho(\hat{G}^D)\rho(\hat{G}^D)^\top]\left(\Theta_\phi^\top\Theta_\omega - \Theta_{\tilde\psi}^\top\Theta_\gamma\right) - \mathbb{E}_{\rho(\hat{G}^D),\epsilon^D}[\rho(\hat{G}^D)\epsilon^D]\}_{D\in\mathcal{D}_{\text{tr}}})) > \dim(\phi) - r$, then $\Theta_\phi^\top\Theta_\omega = \Theta_{\tilde\psi}^\top\Theta_\gamma$ is causal invariant defender for all attack domain set $\mathcal{D}_{\text{all}}$,

According to $Y = C\gamma + \epsilon$, $\tilde\psi(\rho(\hat{G})) = C$, and Step 1, we have

$$\begin{aligned}
&\Theta_\phi\mathbb{E}_{\rho(\hat{G}^D)}\left[\rho(\hat{G}^D)\rho(\hat{G}^D)^\top\right]\Theta_\phi^\top\Theta_\omega\\
&=\Theta_\phi\mathbb{E}_{\rho(\hat{G}^D),Y^D}\left[\rho(\hat{G}^D)Y^D\right]\\
&=\Theta_\phi\mathbb{E}_{\rho(\hat{G}^D),\epsilon^D}\left[\rho(\hat{G}^D)\left(\left(\Theta_{\tilde\psi}\rho(\hat{G}^D)\right)^\top\Theta_\gamma + \epsilon^D\right)\right].
\end{aligned} \tag{17}$$

We can re-write the Eq. 17 as:

$$\Theta_\phi\left(\underbrace{\mathbb{E}_{\rho(\hat{G}^D)}\left[\rho(\hat{G}^D)\rho(\hat{G}^D)^\top\right]\Theta_\phi^\top\Theta_\omega - \mathbb{E}_{\rho(\hat{G}^D),\epsilon^D}\left[\rho(\hat{G}^D)\left(\left(\Theta_{\tilde\psi}\rho(\hat{G}^D)\right)^\top\Theta_\gamma + \epsilon^D\right)\right]}_{:=t_D}\right) = 0 \tag{18}$$

To show that $\Phi$ leads to the desired invariant defender $\Theta_\phi^\top\Theta_\omega = \Theta_{\tilde\psi}^\top\Theta_\gamma$, we assume $\Theta_\phi^\top\Theta_\omega \neq \Theta_{\tilde\psi}^\top\Theta_\gamma$ and reach a contradiction. First, according to Step 2, we have $\dim(\operatorname{span}(\{\mathbb{E}_{\rho(\hat{G}^D)}[\rho(\hat{G}^D)\rho(\hat{G}^D)^\top]\left(\Theta_\phi^\top\Theta_\omega - \Theta_{\tilde\psi}^\top\Theta_\gamma\right) - \mathbb{E}_{\rho(\hat{G}^D),\epsilon^D}[\rho(\hat{G}^D)\epsilon^D]\}_{D\in\mathcal{D}_{\text{tr}}})) > \dim(\phi) - r$. Second, according to Step 1, each $t_D \in \operatorname{Ker}(\phi)$. Therefore, it would follow that $\dim(\operatorname{Ker}(\Theta_\phi)) > \dim(\Theta_\phi) - r$, which contradicts the assumption that $\operatorname{rank}(\Theta_\phi) = r$, which is similar to (Arjovsky et al., 2019). Therefore, $\Phi$ leads to the desired invariant defender $\Theta_\phi^\top\Theta_\omega = \Theta_{\tilde\psi}^\top\Theta_\gamma$.

$\square$

## C Algorithm

In this section, we present the training process for the IDEA algorithm, as illustrated in Algorithm 1. The model $f$ is first optimized using Algorithm 2. During this optimization, the encoder $h$ calculates

---

**Algorithm 1** The training process for IDEA method

---

**Require:** clean graph $G = (\mathcal{V}, \mathcal{E}, X)$, attack method $\Lambda$, set of node labels $Y$
**Ensure:** model $f$ concluding encoder $h$, classifiers $g$ and $g_d$, domain learner $s$
 1: **for** number of training iterations **do**
 2:     Sample minibatch of nodes $V_t$ from node set $\mathcal{V}$, $V_t = \text{Sample}(V)$
      *% Optimize the model $f$*
 3:     Update the model $f$ by Algorithm 2
 4:     Sample minibatch of nodes $V_t$ from node set $\mathcal{V}$, $V_t = \text{Sample}(V)$
      *% Optimize the attack method*
 5:     Generate the perturbed graph by attack method $\Lambda$, $\hat{G} = \Lambda(G)$
 6:     Compute the prediction $\hat{y}$ by the classifier $g$, $\hat{y} = g(z_{\text{atk}})$, where $z_{\text{atk}} = h(\hat{G})[V_t]$
 7:     Compute the attack loss $\mathcal{L}_{\text{atk}} = -\mathcal{L}_{\mathcal{P}}$ , where $\mathcal{L}_{\mathcal{P}}$ is computed by Eq.5
 8:     Compute the gradient of attack method $\Lambda$ and update $\Lambda$.
 9:     Sample minibatch of nodes $V_t$ from node set $\mathcal{V}$, $V_t = \text{Sample}(V)$
      *% Optimize the domain learner*
10:     Generate the perturbed graph by attack method $\Lambda$, $\hat{G} = \Lambda(G)$
11:     Obtain total representation $z$ by concatenating $z_{\text{cln}}$ and $z_{\text{ptb}}$, $z = \text{Concat}(z_{\text{cln}}, z_{\text{ptb}})$, where $z_{\text{cln}} = h(G)[V_t]$ , $z_{\text{ptb}} = h(\hat{G})[V_t]$
12:     Obatin the attack domain $D$ by domain learner $s$, $D = s(z)$
13:     Compute the prediction $\hat{y}$ by the classifier $g$, $\hat{y} = g(z)$
14:     Compute the loss for the domain learner $s$ by Eq.11
15:     Compute the gradient of domain learner $s$ and update $s$
16: **end for**

---

**Algorithm 2** The algorithm of IDEA

---

**Require:** clean graph $G = (\mathcal{V}, \mathcal{E}, X)$, attack method $\Lambda$, set of node labels $Y$, minibatch nodes $V_t$
**Ensure:** updated model $f$
 1: Sample a neighbor for each $v$ in $V_t$ and obtain neighbor nodes $\mathcal{N}_t$, $\mathcal{N}_t = \text{NeighbSample}(V_t)$
 2: Generate the perturbed graph by attack method $\Lambda$, $\hat{G} = \Lambda(G)$
 3: Compute the representation by the encoder $h$ on clean graph $G$ (i.e., $z_{\text{cln}}$) and on perturbed graph $\hat{G}$ (i.e., $z_{\text{ptb}}$), $z_{\text{cln}} = h(G)[V_t]$, $z_{\text{ptb}} = h(\hat{G})[V_t]$
 4: Obtain the total representation $z$ by concatenating $z_{\text{cln}}$ and $z_{\text{ptb}}$, $z = \text{Concat}(z_{\text{cln}}, z_{\text{ptb}})$
 5: Obatin the attack domain $D$ by domain learner $s$, $D = s(z)$
 6: Compute the prediction $\hat{y}$ and prediction based on attack domain $\hat{y}_d$ for nodes $V_t$ by the classifier $g$ and $g_d$, $\hat{y} = g(z)$, $\hat{y}_d = g_d(z, D)$
 7: Compute the predictive loss $\mathcal{L}_{\mathcal{P}}$, node-based invariance loss $\mathcal{L}_{\mathcal{I}}$, and structural-based invariance loss $\mathcal{L}_{\mathcal{E}}$ by Eq.5, Eq.8, and Eq.9, respectively.
 8: Compute the total loss $\mathcal{L} = \mathcal{L}_{\mathcal{P}} + \mathcal{L}_{\mathcal{I}} + \mathcal{L}_{\mathcal{E}}$
 9: Compute the gradient of the model $f$ and update $f$
10: **Return** model $f$

---

the representation $z$ for a minibatch of nodes $V_t$, and the domain learner $s$ identifies the attack domain $D$. Next, classifiers $g$ and $g_d$ produce predictions $\hat{y}$ and $\hat{y}_d$, respectively, which are then used to compute the total loss. After updating $f$, both the attack method and domain learner are optimized. This procedure is repeated iteratively for the number of training iterations.

## D   EXPERIMENTS

### D.1   DATASETS

We conduct node classification experiments on 5 diverse network benchmarks: three citation networks (Cora (Jin et al., 2020b), Citeseer (Jin et al., 2020b), and obgn-arxiv (Hu et al., 2020)), a social network (Reddit (Hamilton et al., 2017; Zeng et al., 2020)), and a product co-purchasing network (ogbn-products (Hu et al., 2020)). Due to the high complexity of some GNN and defense methods, it is difficult to apply them to very large graphs with more than million nodes. Thus, we utilize subgraphs from Reddit and ogbn-products for experiments. Following the settings of most methods (Zügner

Table 3: Statistics of benchmark datasets

| Dataset | Type | #Nodes | #Edges | #Attr. | Classes |
|---|---|---|---|---|---|
| Cora | Citation network | 2,485 | 5,069 | 1,433 | 7 |
| Citeseer | Citation network | 2,110 | 3,668 | 3,703 | 6 |
| Reddit | Social network | 10,004 | 73,512 | 602 | 41 |
| ogbn-products | Co-purchasing network | 10,494 | 38,872 | 100 | 35 |
| ogbn-arxiv | Citation network | 169,343 | 2,484,941 | 128 | 39 |

et al., 2018; Zügner & Günnemann, 2019b; Tao et al., 2021b; Jin et al., 2020b; 2021; Liu et al., 2021; Li et al., 2022c) , experiments are conducted on the largest connected component (LCC). All datasets can be assessed at `https://anonymous.4open.science/r/IDEA_repo-666B`. The statistics of datasets are summarized in Table 3.

- *Cora* (Jin et al., 2020b): A node represents a paper with key words as attributes and paper class as label, and the edge represents the citation relationship.

- *Citeseer* (Jin et al., 2020b): Same as *Cora*.

- *Reddit* (Hamilton et al., 2017; Zeng et al., 2020): Each node represents a post, with word vectors as attributes and community as the label, while each edge represents the post-to-post relationship.

- *ogbn-products* (Hu et al., 2020): A node represents a product sold in Amazon with the word vectors of product descriptions as attributes and the product category as the label, and edges between two products indicate that the products are purchased together.

- *ogbn-arxiv* (Hu et al., 2020): Each node denotes a Computer Science (CS) arXiv paper indexed by (Wang et al., 2020b) with attributes obtained by averaging the embeddings of words in paper's title and abstract. Each edge indicates the citation relationship, and the node label is the primary categories of each arXiv paper.

## D.2 DEFENSE BASELINES

We evaluate the performance of our proposed method, IDEA, by comparing it against ten baseline approaches. These baselines include traditional Graph Neural Networks (GNNs) and defense techniques from three main categories: graph purification, robust aggregation, and adversarial training. For each category, we select the most representative and state-of-the-art methods for comparison. In summary, our comparison includes the following ten baselines:

- Traditional GNNs

  1. *GCN* (Kipf & Welling, 2017): GCN is a popular graph convolutional network based on spectral theory.
  2. *GAT* (Veličković et al., 2018): GAT computes the hidden representations of each node by attending over its neighbors via graph attentional layers.

- Graph purification

  3. *ProGNN* (Jin et al., 2020b): ProGNN simultaneously learns the graph structure and GNN parameters by optimizing three regularizations, i.e., feature smoothness, low-rank and sparsity.
  4. *STABLE* (Li et al., 2022c): STABLE first learns reliable representations of graph structure via unsupervised learning, and then designs an advanced GCN as a downstream classifier to enhance the robustness of GCN.
  5. *GARNET* (Deng et al., 2022): GARNET uses weighted spectral embedding to create a base graph, then refines this graph through the pruning of non-essential edges to enhance adversarial robustness.

- Robust aggregation

  6. *RGCN* (Zhu et al., 2019): RGCN uses gaussian distributions in graph convolutional layers to absorb the effects of adversarial attacks.
  7. *SimPGCN* (Jin et al., 2021): SimPGCN presents a feature similarity preserving aggregation which balances the structure and feature information, and self-learning regularization to capture the feature similarity and dissimilarity between nodes.

8. *Elastic* (Liu et al., 2021): Elastic enhances the local smoothness adaptivity of GNNs via $\ell_1$-based graph smoothing and derives the elastic message passing (EMP).

9. *Soft-Median* (Geisler et al., 2021): Soft-Median is robust aggregation function where the weight for each instance is determined based on the distance to the dimension-wise median.

- Adversarial training

10. *FLAG* (Kong et al., 2022): FLAG, a state-of-the-art adversarial training method, defends against attacks by incorporating adversarial examples into the training set, enabling the model to correctly classify them.

## D.3 ATTACK METHODS

We assess the robustness of IDEA by examining its performance against five adversarial attacks, including one representative poisoning attack (MetaAttack (Zügner & Günnemann, 2019b)) and four evasion attacks (nettack (Zügner et al., 2018), PGD (Madry et al., 2018), TDGIA (Zou et al., 2021), G-NIA (Tao et al., 2021b)). Among these attacks, nettack and MetaAttack modify the original graph structure, while PGD, TDGIA, and G-NIA are node injection attacks. The following is a brief description of each attack:

- *nettack* (Zügner et al., 2018): Nettack is the first adversarial attack on graph data, which can attack node attributes and graph structure with gradient. In this paper, we adopt nettack to attack graph structure, i.e., adding and removing edges.

- *PGD* (Madry et al., 2018): PGD, a popular adversarial attack, is used as node injection attack. We employi projected gradient descent (PGD) to inject malicious nodes on graphs.

- *TDGIA* (Zou et al., 2021): TDGIA consists of two modules: the heuristic topological defective edge selection for injecting nodes and smooth adversarial optimization for generating features of injected nodes.

- *G-NIA* (Tao et al., 2021b): G-NIA is one of the state-of-the-art node injection attack methods, showing excellent attack performance. G-NIA models the optimization process via a parametric model to preserve the learned attack strategy and reuse it when inferring.

- *MetaAttack* (Zügner & Günnemann, 2019b): MetaAttack is the most representative poisoning attack method, which has been widely-used to evaluate the robustness of GNN models.

## D.4 IMPLEMENTATION DETAILS

For attack and defense methods, we employ the most widely recognized DeepRobust (Li et al., 2021) benchmark in the field of graph adversarial and defense, to ensure that the experimental results can be compared directly to other papers that use DeepRobust (such as Elastic (Liu et al., 2021), ProGNN (Jin et al., 2020b), STABLE (Li et al., 2022c), and SimPGCN (Jin et al., 2021)). For our IDEA, hyper-parameters of all datasets can be assessed at `https://anonymous.4open.science/r/IDEA_repo-666B`. Note that we implement both attribute and structural attacks to generate adversarial examples that minimize the predictive loss $\mathcal{L}_\mathcal{P}$. Specifically, attribute attack generation is the same as that in FLAG (Kong et al., 2022), while structural attack generation is the same as that in EERM (Wu et al., 2022). For all methods that require a backbone model (e.g., FLAG and our IDEA), we use GCN as the backbone model. All experiments are conducted on a single NVIDIA V100 32 GB GPU.

## D.5 HYPER-PARAMETER ANALYSIS

We investigate the effects of coefficient $\alpha$ and the number of domains and compare the defense performance. Note that we take results against evasion attacks on Cora as an illustration. Figure 6 shows that the average accuarcy of the clean and attacked graphs, along with the standard deviation of accuracy across these graphs, i.e., AVG in Section 4.1. For coefficient $\alpha$, we observe that when $\alpha$ is increasing, IDEA achieves better performance (higher accuracy), and performs more stable (lower standard deviation), validating the effectiveness of invariance component. While, too large $\alpha$ (e.g. $\alpha = 150$) causes domination of invariance goal, leading to little attention to the predictive goal and degradation of performance. Regarding the number of attack domains, performance improves with

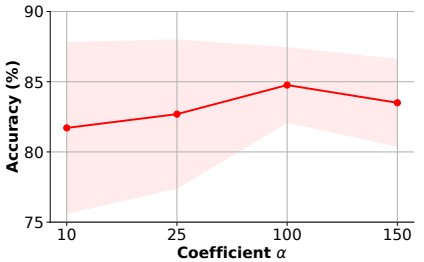 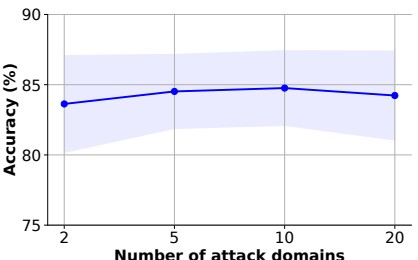

Figure 6: Hyperparameter analysis: The average accuracy of clean and attacked graphs, including Clean, nettack, PGD, TDGIA, and G-NIA.

Table 4: Accuracy(%) of targets under adaptive attack.

| Dataset | GCN | ProGNN | STABLE | RGCN | SimPGCN | FLAG | IDEA |
|---------|-----|--------|--------|------|---------|------|------|
| Cora | 18.7±3.5 | 15.0±2.8 | 27.5±5.0 | 14.3±1.6 | 28.9±3.4 | 36.7±2.4 | **53.1±5.0** |
| Citeseer | 11.8±2.0 | 21.8±2.3 | 12.7±2.3 | 10.1±1.2 | 24.5±4.7 | 31.1±6.4 | **44.4±1.6** |
| Reddit | 34.7±4.9 | 43.1±8.1 | 27.3±4.4 | 57.5±3.0 | 12.5±6.7 | 5.2±5.9 | **61.7±5.3** |

increasing domain numbers, reaching its peak at 10 domains. This may be due to the relatively small number of nodes in the Cora dataset, suggesting that a larger number of domains (e.g., 20) is not necessary. In our main experiment shown in Table 1, we utilized $\alpha = 100$ and the attack domain number to 10 to achieve the best results.

### D.6 Performance under Adaptive Attacks

To better evaluate our IDEA, we also conduct experiments under adaptive attack, i.e., PGD in (Mujkanovic et al., 2022). We implement adaptive attacks for baselines and IDEA. Some baselines are excluded because their open source codes use edge_index to represent edges. This makes calculating gradients on edges challenging, so the implementation of these baselines are difficult to conduct white-box adaptive attacks. As shown in Table 4, adaptive attack causes serious performance degradation to defense methods because adaptive attacks are powerful white-box attacks. IDEA outperforms all the baselines. Experiments offer a more broader evaluation of IDEA's performance under a hard scenario, consistently showing IDEA's superiority.

### E Symbol and Definition

Table 5 summarizes all symbols and their definitions for quick reference.

Table 5: Symbol table

| Symbol | Definition |
|---|---|
| $G$ | Graph in a node classification task |
| $\mathcal{V}$ | Node set of a graph |
| $\mathcal{E}$ | Edge set of a graph |
| $X$ | Attribute matrix |
| $\mathcal{K}$ | Class set |
| $K$ | Class number |
| $f_\theta$ | GNN classifier |
| $\hat{G}$ | Perturbed graph |
| $\mathcal{G}$ | Admissible perturbed graph set |
| $i, j$ | Nodes in graph |
| $G_i$ | Input ego-network of node $i$ |
| $Y_i$ | Label of node $i$ |
| $C_i$ | Causal feature of node $i$ |
| $D_i$ | Attack domain of node $i$ |
| $N_i$ | Non-causal feature of node $i$ |
| $I(\cdot)$ | Mutual information |
| $\Phi$ | Feature encoder |
| $C_\mathcal{N}$ | Causal feature of neighbor $\mathcal{N}$ |
| $Z$ | Representation of feature encoder |
| $p(\cdot)$ | Natural distribution |
| $q(\cdot)$ | Variation approximation |
| $h$ | Neural network feature encoder |
| $g$ | Neural network classifier |
| $g_d$ | Neural network auxiliary classifier |
| $s$ | Domain learner |
| $\mathcal{V}^D$ | Nodes assigned to domain $D$ |
| $r^D$ | Overall representation of $\mathcal{V}^D$ |
| $\mathcal{L}_\mathcal{P}$ | Predictive loss |
| $\mathcal{L}_\mathcal{I}$ | Node-based invariance loss |
| $\mathcal{L}_\mathcal{E}$ | Structure-based invariance loss |
| $\mathcal{L}_\mathcal{D}$ | Domain loss |
| $\gamma$ | Intrinsic causal mechanism |
| $\epsilon$ | Gaussian noise |
| $\psi$ | Mapping from causal and non-causal features to graph representation |
| $\rho$ | Powerful graph representation extractor |
| $\Theta.$ | Parameter associated with a particular model component |

