# OpenReview forum: "IDEA: Invariant Causal Defense for Graph Adversarial Robustness"
_ICLR.cc/2024/Conference — Submitted to ICLR 2024_

### Official Review · Reviewer_P55P · 2023-10-25

**Soundness:** 2 fair
**Presentation:** 2 fair
**Contribution:** 2 fair
**Rating:** 6
**Confidence:** 3

**Summary:**

The paper introduces a method to disentangle causal and non causal features in order to protect a gnn from adversarial attacks

**Strengths:**

- Good presentation
- Great experimentation
- The addition of the ablation study is very welcome
- Interesting premise and motivation
- Interesting approach

**Weaknesses:**

- Minor language issues throughout the paper
- It is not clear why the end result of the encoder H would be a causal feature. I can see how the learned representation could be a feature set that is invariant to the features the attack but making a causal claim is not completely justified.
- it is unclear how the method generalises to multiple unseen attacks

**Questions:**

- why is the features learned causal and not just invariant to the attack ?
- how does the method generalise to types of attacks not seen in training ?


EDIT AFTER REBUTTAL

updated score from 5 to 6

---

> ### Author Response · Authors · 2023-11-14
> **Response to Reviewer P55P**
>
> Dear Reviewer P55P,
>
> Thank you for recognizing the strengths of our work, including our interesting motivation, interesting approach, good representation, and great experimentation. We appreciate the opportunity to provide a detailed response to the weaknesses and questions you have highlighted.
>
> **Q1:** Minor language issues throughout the paper
>
> **A1:** We have thoroughly revised the paper to ensure that the language is clear, precise, and adheres to high academic standards. For ease of identification, all modifications have been highlighted in blue in the revised version of our paper.
>
> **Q2:** It is not clear why the end result of the encoder H would be a causal feature. I can see how the learned representation could be a feature set that is invariant to the features the attack but making a causal claim is not completely justified.
>
> **A2:** We appreciate your recognition that the learned representation is invariant to attacks. We would like to offer further clarification on our use of the term "causal feature." This terminology is specifically chosen to denote features that maintain a direct and invariant causal relationship with the labels, as supported by the principles found in the invariant causal learning literature [1,2,3,4]. For instance, the concept of Invariant Risk Minimization (IRM) posits that invariance is a defining characteristic of causal relationships [1].
>
> In our initial paper, detailed in Section 3.1 and Section 3.2.1, we delineate our intention to identify features that hold a causal influence over the labels. By examining causal relationships and conditional independencies, we have developed the predictive and invariant goals. These goals are designed to steer the encoder $h$ towards learning these causal features, as further elaborated in Section 3.2.2.
>
> We trust that this elaboration dispels any ambiguity surrounding our use of the term "causal".
>
> **Q3:** It is unclear how the method generalises to multiple unseen attacks.
>
> **A3:** As stated in Section 3.2, our IDEA aims to learn the causal features that determine labels, having strong and invariant predictability across different attack domains, rather than spurious relationships that are limited to a single domain. As stated in Section 3.2.1, the element vulnerable to the attack, denoted as $G$ (since the attack propagates from attack domain $D$ to perturbation $T$ and subsequently to ego-network $G$), exhibits certain characteristics; specifically, the conditional mutual information $I(D,Y|G)$, is substantially high, because collider. Our invariance goal is to minimize the conditional mutual information $I(D,Y|Z)$, avoiding $Z$ learning the part affected by the attack. Meanwhile, our component of domain learner aids in proving IDEA achieving adversarial robustness across all attack domains, as proved in Proposition 2.
>
> In addition to theoretical guarantees, experimental results further demonstrate that the IDEA is robust to attacks that do not appear in the training set, thereby confirming the effectiveness and generalization of our method against unseen attacks, and this strength is also identified by Reviewer jN54.
>
> In conclusion, we greatly appreciate your recognition of our method and experimentation. We observe that your concerns are mainly regarding the conception, which has been stated in our initial paper. We have provided further clarification to address each concern. We hope you can reconsider your assessment and potentially improve the overall assessment based on our clarification. We believe that our work offers valuable insights and advances the state-of-the-art in adversarial graph machine learning, making a meaningful contribution to the research community. We appreciate your valuable feedback, and we look forward to any further comments or questions you may have. Thank you for your consideration.
>
> Best regards,
>
> The authors
>
> [1] Martin Arjovsky, Léon Bottou, Ishaan Gulrajani, and David Lopez-Paz. Invariant risk minimization.
>
> [2] Bo Li, Yifei Shen, Yezhen Wang, Wenzhen Zhu, Dongsheng Li, Kurt Keutzer, and Han Zhao. Invariant information bottleneck for domain generalization. In AAAI 2022.
>
> [3] Qibing Ren, Yiting Chen, Yichuan Mo, Qitian Wu, and Junchi Yan. Dice: Domain-attack invariant causal learning for improved data privacy protection and adversarial robustness. In KDD 2022
>
> [4] Yongqiang Chen, Yonggang Zhang, Yatao Bian, Han Yang, MA KAILI, Binghui Xie, Tongliang Liu, Bo Han, and James Cheng. Learning causally invariant representations for out-of-distribution generalization on graphs. In NeurIPS 2022.

---

> ### Author Response · Authors · 2023-11-17
> **Followup on our response to Reviewer P55P**
>
> Dear Reviewer P55P,
>
> We are respectfully writing to follow up on our previous response with regard to the review of our paper. In response to your valuable comments, we have provided detailed clarification to address each concern. We have noticed that your concerns are mainly regarding the conception, which has been stated in our initial paper. We sincerely hope that our clarifications can address your concerns.
>
> We eagerly await your feedback and are ready to offer any further clarifications or information you may require.
>
> Thank you once again for your recognition and contributions to our work.
>
> Best regards,
>
> The authors

---

> > ### Comment · Reviewer_P55P · 2023-11-17
> > **Acknowledgement of Rebuttal**
> >
> > I acknowledge the rebuttal of the authors, I am somewhat satisfied with the responses provided, as such in the interest of discussion in the community i will slightly increase my score.

---

> > > ### Author Response · Authors · 2023-11-17
> > > **Thank you for increasing your rating**
> > >
> > > Dear Reviewer P55P,
> > >
> > > Thank you for increasing your rating!
> > >
> > > We are greatly encouraged by your satisfaction with our response. We appreciate your recognition of the benefits and potential of our work in the community.
> > >
> > > If any aspect remains unclear or you have further queries, please feel free to let us know. We are fully prepared to provide additional clarification or information as needed. Thank you once again for your recognition and contributions to our work.
> > >
> > > Best regards,
> > >
> > > The authors

---

### Official Review · Reviewer_jN54 · 2023-10-30

**Soundness:** 3 good
**Presentation:** 3 good
**Contribution:** 2 fair
**Rating:** 8
**Confidence:** 2

**Summary:**

This paper uses a new causal defense perspective to resist adversarial attacks by learning powerful and invariant predictable causal features, and proposes the Invariant causal defense method against adversarial attacks (IDEA). Experiments have proven that this method is effective against various attack methods and has excellent performance and strong generalization.

**Strengths:**

1. The article contains a more complete proof process and theoretical basis.

2. The article learns powerful and immutable causal features to deal with adversarial attacks from a relatively novel causal defense perspective.

3. The experiment proves the effectiveness and generalization of the method proposed in the paper.

**Weaknesses:**

1. Judging from the results shown in Table 1 and Table 2, compared with other denoising methods, the performance of this method has indeed been greatly improved. However, the table only shows the excellence of this method when facing one of poisoning attacks or evasion attacks, and the results for other attack methods are not shown.

2. There are many symbols listed in the article, and it seems unclear when mixed together.

3.The drawing of the overall block diagram of the method is relatively rough.


============================================================
After rebuttal

The authors solve most of my concerns. Thus, I am willing to increase the rating score from 6 to 8.

**Questions:**

1. Does this method still have such obvious advantages in the face of the other attack methods mentioned in the article?

2. There are also some purification methods that seem to be able to be extended to graph purification. Whether they will also encounter the limitations mentioned by the author, I hope the author can discuss or compare this. Such as the following methods:

[1] Shi C, Holtz C, Mishne G. Online adversarial purification based on self-supervision[J]. arXiv preprint arXiv:2101.09387, 2021.

[2] Liao F, Liang M, Dong Y, et al. Defense against adversarial attacks using high-level representation guided denoiser[C]//Proceedings of the IEEE conference on computer vision and pattern recognition. 2018: 1778-1787.

[3] Zhou D, Wang N, Peng C, et al. Removing adversarial noise in class activation feature space[C]//Proceedings of the IEEE/CVF International Conference on Computer Vision. 2021: 7878-7887.

[4] Naseer M, Khan S, Hayat M, et al. A self-supervised approach for adversarial robustness[C]//Proceedings of the IEEE/CVF Conference on Computer Vision and Pattern Recognition. 2020: 262-271.

---

> ### Author Response · Authors · 2023-11-14
> **Response to Reviewer jN54 (1/2)**
>
> Dear Reviewer jN54,
>
> Thank you for recognizing the strengths of our paper, including the comprehensive theoretical basis, the novelty of perspective, as well as the effectiveness and generalization of our method as demonstrated by our experiments. Regarding the weaknesses you have identified, we provided explanations and supplemented them where appropriate in the paper, as follows:
>
> **Q1:** The table only shows the excellence of this method when facing one of poisoning attacks or evasion attacks.
>
> **A1:**  We would like to clarify that there seems to be a misunderstanding regarding the scope of the attacks tested. In fact, our IDEA has been rigorously tested against four evasion attacks. These include the graph modification attack Nettack and node injection attacks including PGD, TDGIA, and G-NIA. In addition to evasion attacks, we have also employed the widely acknowledged metattack for poisoning scenarios, testing it across different perturbation rates. This approach is a standard in defense-related research, as evidenced by its extensive use in prior studies [1,2,3,4]. The results in our initial submission broadly cover various types of attacks, including untargeted (MetaAttack) and targeted (nettack, PGD, TDGIA, and G-NIA) attacks, graph modification attacks (nettack and MetaAttack) and node injection attacks (PGD, TDGIA, and G-NIA), as well as white-box attacks (adaptive attack in Table 4) and gray-box (MetaAttack, nettack, PGD, TDGIA, and G-NIA) attacks. These have been stated in “implementation” in Section 4 in our initial submission.
>
> **Q2:** The symbols seem unclear.
>
> **A2:** We appreciate your valuable feedback. To address this, we have revised the manuscript to include Table 5 that summarizes all symbols and their definitions for quick reference in Appendix E. We believe this can make the paper more reader-friendly and help to ensure that the symbols are clear and well-understood.
>
> **Q3:** The drawing of overall block diagram of the method.
>
> **A3:** Thank you for your constructive feedback. In response, we have further improved Figure 3, which illustrates the overall architecture, enhancing its layout and alignment. Please refer to the updated paper for a detailed view.  Furthermore, we have appended a description to Figure 3, aiming to augment its clarity and facilitate better understanding for the readers. Thank you for bringing this to our attention.
>
> **Q4:** Can purification methods in CV extend to graph purification?
>
> **A4:** We read the computer vision purification literature you recommended [5, 6, 7, 8] and observed that they often leverage image data priors. For example, paper [5] incorporates self-supervised tasks such as image rotation that are unique to the domain of computer vision, while paper [8] depends on a pixel loss function, i.e., $\mathcal{L}_{img}$, to encourage image smoothness. These domain-specific dependencies pose significant challenges when considering the direct transposition of these methods to graph data, which inherently lacks such image-based priors.
>
> In contrast, graph purification methods are specifically designed to exploit the unique structural and feature-related properties of graph data, making them appropriate for addressing graph-specific issues. However, graph purification defenses rely on heuristic assumptions, as detailed in our initial submission. Experimental results demonstrate that our method IDEA outperforms these existing methods when defending against attacks.
>
>  In our revised paper, we have added a separate subsection Appendix A.4 discussing these purification methods in the computer vision field [5, 6, 7, 8]. We believe this extended discussion will provide our readers with a more holistic understanding of the challenges and potential of purification methods across various fields.
>
> In conclusion, we are encouraged by your recognition on our novelty and effectiveness. To address your concerns, we have implemented clarifications and revisions. These include the addition of a symbol table for quick reference, the refinement of Figure 3 for enhanced clarity, and the inclusion of a new subsection, Appendix A.4, which expands on related work and discussion. We trust that these updates will provide a clearer understanding of our work. We kindly request that you re-evaluate our submission in light of these modifications. We believe that our work offers valuable insights and advances the state-of-the-art in adversarial graph machine learning, making a meaningful contribution to the research community. We are grateful for your constructive feedback and are open to any additional comments or inquiries you may have. Thank you for your consideration.
>
> Best regards,
>
> The authors

---

> ### Author Response · Authors · 2023-11-14
> **Response to Reviewer jN54 (2/2)**
>
> [1] Wei Jin, Yao Ma, Xiaorui Liu, Xian-Feng Tang, Suhang Wang, and Jiliang Tang. Graph structure learning for robust graph neural networks. In KDD 2021
>
> [2] Wei Jin, Tyler Derr, Yiqi Wang, Yao Ma, Zitao Liu, and Jiliang Tang. Node similarity preserving graph convolutional networks. In WSDM 2021
>
> [3] Kuan Li, Yang Liu, Xiang Ao, Jianfeng Chi, Jinghua Feng, Hao Yang, and Qing He. Reliable representations make a stronger defender: Unsupervised structure reﬁnement for robust gnn. In KDD 2022
>
> [4] Xiaorui Liu, Wei Jin, Yao Ma, Yaxin Li, Hua Liu, Yiqi Wang, Ming Yan, and Jiliang Tang. Elastic graph neural networks. In ICML 2021
>
> [5] Shi C, Holtz C, Mishne G. Online adversarial purification based on self-supervision[J]. arXiv preprint arXiv:2101.09387, 2021.
>
> [6] Liao F, Liang M, Dong Y, et al. Defense against adversarial attacks using high-level representation guided denoiser[C]//Proceedings of the IEEE conference on computer vision and pattern recognition. 2018: 1778-1787.
>
> [7] Zhou D, Wang N, Peng C, et al. Removing adversarial noise in class activation feature space[C]//Proceedings of the IEEE/CVF International Conference on Computer Vision. 2021: 7878-7887.
>
> [8] Naseer M, Khan S, Hayat M, et al. A self-supervised approach for adversarial robustness[C]//Proceedings of the IEEE/CVF Conference on Computer Vision and Pattern Recognition. 2020: 262-271.

---

> ### Author Response · Authors · 2023-11-17
> **Followup on response to Reviewer jN54**
>
> Dear Reviewer jN54,
>
> We are respectfully writing to follow up on our previous response with regard to the review of our paper. In response to your valuable comments, we have provided detail clarification to address each concern. We have noticed that there may be some misunderstanding on the scope of the attacks tested, which has been detailed in Section 4. We also have provided: the addition of a symbol table for quick reference, the refinement of Figure 3 for enhanced clarity, and the inclusion of a new subsection Appendix A.4, which expands on related work and discussion on CV purification methods. We sincerely hope that our clarifications and revisions can address your concerns.
>
> We eagerly await your feedback and are ready to offer any further clarifications or information you may require.
>
> Thank you once again for your recognition and contributions to our work.
>
> Best regards,
>
> The authors

---

> ### Comment · Reviewer_jN54 · 2023-11-17
> **Response to authors**
>
> Dear authors,
>
> Thanks for your answers to my confusions. I believe that your revisions on symbols and figures can enhance the readability and quality of the paper. In addition, the discussion and analysis of purification methods between visual data and graph data are also provided, which I think can provide readers with a more comprehensive understanding of purification strategies. Based on these, I am willing to increase the rating score.

---

> > ### Author Response · Authors · 2023-11-17
> > **Thank you for increasing your rating from 6 to 8**
> >
> > Dear Reviewer jN54,
> >
> > Thank you for increasing your rating from 6 to 8!
> >
> > We are greatly encouraged that we have solved your concerns. We appreciate your recognition of the improved readability and the comprehensive discussion of purification methods in our revised paper.
> >
> > In addition, we are fully prepared to provide any additional clarification or information as needed. Thank you once again for your recognition and contributions to our work.
> >
> > Best regards,
> >
> > The authors

---

### Official Review · Reviewer_6toz · 2023-11-01

**Soundness:** 3 good
**Presentation:** 3 good
**Contribution:** 2 fair
**Rating:** 6
**Confidence:** 3

**Summary:**

The manuscript proposes a new framework for adversarial robustness in GNNs. The primary subject of interest is the learning of causal features defending against evasion and poisoning attacks. The empirical results are further supported by theoretical analyses with provable defense guarantees.

**Strengths:**

Overall the experimental methodology is sound with complete theoretical derivations.

**Weaknesses:**

The link is expired (https://anonymous.4open.science/r/IDEA_repo-666B), which made further investigation on code artifact and validating empirical results hard. Therefore, the claims made in the paper cannot be carefully checked. Additionally, the claim that causality directly contributes to improved defense performance is weak, as opposed to algorithmic superiority.

**Questions:**

What's the significance of Figure 1 (b)? Aren't there inherent learning capability differences between different GNN architectures?

---

> ### Author Response · Authors · 2023-11-14
> **Response to Reviewer 6toz**
>
> Dear Reviewer 6toz,
>
> Thank you for recognizing the strengths of our sound method and complete theoretical derivations. We understand your concerns and would like to address them as follows:
>
> **Q1:** The expired link of code.
>
> **A1:** We sincerely apologize for the inconvenience caused by the expired link. The issue has been resolved, and the repository is now available at the updated link: https://anonymous.4open.science/r/IDEA_repo-666B. We have ensured that this repository contains all the necessary code, datasets, and hyperparameters for comprehensive validation of our empirical results. Additionally, we submitted all these resources in the "Supplementary Material" on our initial submission on the OpenReview website, for complete transparency and ease of reproducibility. We trust that the updated link and these resources can address any concerns regarding the validation of our work.
>
> **Q2:** The claim that causality directly contributes to improved defense performance is weak, as opposed to algorithmic superiority.
>
> **A2:** We respectfully disagree with this assertion. In our paper, we have not claimed that "causality directly contributes to improved defense performance". But, the causality is important, since the goals of IDEA are obtained based on the causality. We have explained the relationship between the causality and IDEA’s goal in Section 3.2.1 in our initial submission. Here, we provide the summary:
>
> - Predictive goal $\max_{\Phi} I\left(\Phi(G), Y\right)$ is according to causality $C_i \rightarrow Y_i$.
> - Node-based Invariance goal $\min_{\Phi}  I\left(Y, D \mid \Phi(G)\right)$ is derived by comparing the conditional independences $ Y_i
> \not\negthickspace \perp \negthickspace\negthickspace \perp D_i \mid G_i$ and $Y_i \perp \negthickspace\negthickspace \perp D_i \mid C_i$, which are based on causalities in Figure 2.
> - Structural-based Invariance goal $\min_{\Phi}  I\left(Y, D \mid \Phi(G)\_\mathcal{N}\right)$ is derived by comparing $Y_{i} \not\negthickspace \perp \negthickspace\negthickspace \perp D_{i} \mid G_{k}$ and $Y_{i} \perp \negthickspace\negthickspace \perp D_{i} \mid C_{k}$, which are based on causalities in Figures 2.
>
> **Q3:** What's the significance of Figure 1 (b)? Aren't there inherent learning capability differences between different GNN architectures?
>
> **A3:** The purpose of this figure is to demonstrate how various assumptions influence clean performance, as stated in our initial Section 1. Note that the basic architecture (such as the hidden layer dimensions and a number of hidden layers) of defense methods (such as ProGNN, STABLE, and SimpGCN) are consistent with GCN, to ensure the comparison is fair.
>
> Despite their architectural similarities, we observe a notable degradation in the performance of ProGNN, STABLE, and SimpGCN when they process clean graphs. This observation is critical as it highlights the limitations of these models. For example, ProGNN aims to counteract adversarial attacks by modifying the graph structure; however, when applied to clean graphs, this strategy may inadvertently remove legitimate edges, thereby impairing the model's performance.
>
> In essence, Figure 1(b) shows the influence of these heuristic priors on clean performance, highlighting the limitation of existing methods.
>
>  We observe that some of your concerns come from some misunderstanding which has been stated in our initial paper. We have provided further clarification to address each concern and updated the repository to strengthen the validation of our paper's claims. We hope you can reconsider your assessment and potentially improve the overall assessment. We believe that our work offers valuable insights and advances the state-of-the-art in adversarial graph machine learning, making a meaningful contribution to the research community. We appreciate your valuable feedback, and we look forward to any further comments or questions you may have. Thank you for your consideration.
>
> Best regards,
>
> The authors

---

> > ### Author Response · Authors · 2023-11-21
> > **Follow-up: Discussion deadline approaching**
> >
> > Dear Reviewer 6toz,
> >
> > With the discussion deadline approaching, we wish to respectfully remind you of the pending feedback for our paper. While we have received responses from other reviewers to our clarifications, which have been greatly appreciated, we have not received your valuable input on our rebuttal and the additional materials we have submitted.
> >
> > Your insights are very important to us, and we are keen to hear your thoughts on our clarifications and updated repository.  We eagerly await your feedback and are ready to offer any further clarifications or information you may require.
> >
> > Thank you once again for your recognition and contributions to our work.
> >
> > Best regards,
> >
> > The authors

---

> > > ### Comment · Reviewer_6toz · 2023-11-21
> > >
> > > Thank you for your comments, I think A1-3 are satisfactory. I will adjust my score accordingly.

---

> > > > ### Author Response · Authors · 2023-11-21
> > > > **Thank you for increasing your rating**
> > > >
> > > > Dear Reviewer 6toz,
> > > >
> > > > Thank you for increasing your rating!
> > > >
> > > > We are greatly encouraged by your satisfaction with our response. If any aspect remains unclear or you have further queries, please feel free to let us know. We are fully prepared to provide additional clarification or information as needed. Thank you once again for your recognition and contributions to our work.
> > > >
> > > > Best regards,
> > > >
> > > > The authors

---

> ### Author Response · Authors · 2023-11-17
> **Followup on response to Reviewer 6toz**
>
> Dear Reviewer 6toz,
>
> We are respectfully writing to follow up on our previous response with regard to the review of our paper. In response to your valuable comments, we have provided detailed clarification to address each concern and updated the repository to strengthen the validation of our paper's claims. We have noticed that there may be some misunderstandings on the significance of causality, which have been stated in Section 3.2.1. We sincerely hope that our clarifications and the updated repository can address your concerns.
>
> We eagerly await your feedback and are ready to offer any further clarifications or information you may require.
>
> Thank you once again for your recognition and contributions to our work.
>
> Best regards,
>
> The authors

---

### Official Review · Reviewer_knCN · 2023-11-07

**Soundness:** 2 fair
**Presentation:** 3 good
**Contribution:** 3 good
**Rating:** 5
**Confidence:** 2

**Summary:**

The paper proposes a casual defense to improve the graph adversarial robustness. Specifically, it first defines a causal graph that some casual feature would have strong predictability for the label and maintains invariant predictability across attack domains so that perturbing the features adversarially won't induce a successful attack. It then defines a objective by defining different mutual information to learn the casual features. The experiments show the proposed method could achieve a significant improvement over current defenses.

**Strengths:**

1. The paper provides a new perspective on the casual inference to defend against graph adversarial attacks.
2. The proposed method shows significant improvement on both evasion and poisoning settings.

**Weaknesses:**

1. Since I am not expert in causal inference, it is unclear to me how the initial casual graph is defined. And I am not clear whether the graph is based on author's assumption or derived automatically. If it is the former case, the truthfulness of the provided causal graph is debatable.
2. The threat model is actually unclear. The proposed method actually built a detector neural network in the defense. The attacks tested seems have no knowledge about the detector network. Therefore, the improvement might be brought by the attacker's incapability acquiring enough model information. Also,   it is unfair to compare with other proposed methods since they are only modifying the provided model or aggression rule. A adaptive attack  or white box attack should assume the attacker has already known the added detection neural networks.
3. There are some notations and figures problems that causes the paper not easy to follow.
Node j in Figure 2 should be Node k.  Z in Section 3.2.1 is not defined.  (.)_\cN is never defined. The overall framework only shows in Figure 3 without any introduction. Empty graph in Figure 5.

**Questions:**

1. Why does causal feature would only connect with label and input data? Is it defined or derived just based on some assumption the paper made or is there anyway to automatic define the graph?
2. If the attacker knew the added neural network, would the proposed method still achieve a similar improvement in the paper?

---

> ### Author Response · Authors · 2023-11-14
> **Response to Reviewer knCN (1/2)**
>
> Dear Reviewer knCN,
>
> Thank you for acknowledging our new perspective and the significant improvement of our method on experiments. We appreciate the opportunity to clarify the points you have raised and hope the following responses will enhance your understanding of our work.
>
> **Q1:** Since I am not expert in causal inference, it is unclear to me how the initial casual graph is defined.
>
> **A1:** Thank you for your question. Our interaction causal model is grounded in the well-established understanding in the field of invariant causal learning [1,2,3] and our analysis of graph adversarial attacks. We have elucidated the explanation of each causal relationship in Section 3.1, and provide a summary below:
>
> 1. The causal feature $C$ determines the input ego-network $G$ and label $Y$. According to the literature [1,2,3], the causes of input data generation are divided into two categories for simplicity. (i) The causes that causally determine the label are called causal feature $C$, leading to the causal relationships $C \rightarrow Y$ and $C \rightarrow G$. For example, in credit scoring, $C$ represents the financial situation, which determines both $G$ (including personal attributes and friendships) and credit score $Y$. (ii) The causes that do not influence labels are called non-causal features $N$. These are recognized in the causal invariant field [1,2,3].
> 2. In the graph adversarial field, the attack domain $D$ influences the perturbation $T$, and the perturbation $T$ influences the input ego-network $G$. We summarize $D$ and $T$ as the non-causal features $N$.
> 3. Particularly, in graph data, due to the interconnectedness of nodes, we introduce the interaction causal model to capture this non-IID characteristic. The causal feature $C_i$ and perturbation $T_i$ of a node $i$ affect its neighbor $G_k$, which is also consistent with the GNN literature [4,5].
>
> **Q2:** The knowledge of the threat model and “detector network”
>
> **A2:** We would like to address the misunderstandings regarding the threat model and our method:
>
> - **We have provided adaptive attack results**: In our initial submission, we have indeed included the evaluation against white-box adaptive attacks [9], as detailed in Appendix D.4 titled "Performance under adaptive attacks". Here, the adaptive attacker is fully aware of all the information of defense methods, including the architecture, parameters, as well as input and output data. As presented in Table 4, the results demonstrate that while adaptive attacks significantly challenge defense methods, our proposed method IDEA maintains superior performance over all baseline methods. This rigorously tests IDEA's robustness in scenarios where the attacker has complete knowledge of the defense strategy.
> - **Details on the threat model are in Section 4**: We have provided the information of the attack in the "Implementation" in Section 4 of our initial paper, found on page 7. "Implementation" outlines the attack phase (evasion/poisoning), target nodes, and knowledge of the attacker.
> - **Misunderstanding of our method**: We would like to rectify a misunderstanding regarding our proposed method. IDEA does not employ a separate 'detector network' as part of its defense strategy. Instead, it focuses on learning causal features through the optimization of node-based and structure-based invariance losses that we have proposed. This approach is fundamentally different from detection-based defenses and is integral to the novelty of our contribution.
> - **Fair comparison**: We assure you that all comparative evaluations have been conducted under the same conditions. Both IDEA and the baseline methods are provided with the same level of information regarding the attacks they are tested against. This ensures that the comparisons are fair and the improvements observed with IDEA are attributable to its inherent robustness and effectiveness.
>
> **Q3:** Notation and figure issues
>
> **A3:** We apologize for any confusion stemming from inaccuracies in notations and figures previously presented. In the updated version of the paper,
>
> - Node $j$ in Figure 2 is now corrected as node $k$.
> - The notation $Z$ in Section 3.2.1 has been updated to $\Phi(G)$ for clarity.
> - The symbol $\mathcal{N}$, denoting neighbors, has been clarified.
>
> There are also some misunderstandings about our paper in the review comments:
> - Framework introduction: In our initial submission, we have provided the introduction of the framework in the 3.2.4 “Overall Framework” on page 6.
> - Figure 5 viewing issue: This issue may be attributed to browser-specific rendering limitations. We kindly recommend downloading the PDF version of our paper and viewing it with a PDF reader for accurate display.
>
> We have meticulously revised the entire document to prevent any such errors in the paper. For ease of identification, all modifications have been highlighted in blue in the revised version of our paper.

---

> ### Author Response · Authors · 2023-11-14
> **Response to Reviewer knCN (2/2)**
>
> **Q4:** Why does causal feature would only connect with label and input data?
>
> **A4:** We respectfully contest the assertion that "causal feature would only connect with the label and input data." Contrarily, we introduce causal feature as an abstraction of underlying factors that determine both inputs and labels. This is consistent with the existing causal invariance literature [1,2,3].
>
> **Q5:** If the attacker knew the added neural network, would the proposed method still achieve a similar improvement in the paper?
>
> **A5:** We have already conducted the white-box adaptive attack [9] experiments, and exported the results in Appendix D.4 “Performance under adaptive attacks”. As shown in Table 4, adaptive attacks make serious performance degradation to defense methods because they are powerful white-box attacks. IDEA outperforms all the baselines. The results offer a broader evaluation of IDEA’s performance under a hard scenario, consistently showing IDEA’s superiority.
>
> We observe that there are quite a lot of misunderstandings on our paper and most of them have been stated in our initial submission. We have provided further clarification to address each concern. We hope you can reconsider your assessment and potentially improve the overall assessment based on our clarification. We believe that our work offers valuable insights and advances the state-of-the-art in adversarial graph machine learning, making a meaningful contribution to the research community. We are available for any further clarifications you may require. Thank you for your consideration.
>
> Best regards,
>
> The authors
>
> [1] Yonggang Zhang, Mingming Gong, Tongliang Liu, Gang Niu, Xinmei Tian, Bo Han, Bernhard Schölkopf, and Kun Zhang. Adversarial robustness through the lens of causality. In ICLR 2022.
>
> [2] Qibing Ren, Yiting Chen, Yichuan Mo, Qitian Wu, and Junchi Yan. Dice: Domain-attack invariant causal learning for improved data privacy protection and adversarial robustness. In KDD 2022.
>
> [3] Bo Li, Yifei Shen, Yezhen Wang, Wenzhen Zhu, Dongsheng Li, Kurt Keutzer, and Han Zhao. Invariant information bottleneck for domain generalization. In AAAI 2022.
>
> [4] Petar Veliˇckovi´c, Guillem Cucurull, Arantxa Casanova, Adriana Romero, Pietro Liò, and Yoshua Bengio. Graph Attention Networks. In ICLR 2018
>
> [5] Thomas N. Kipf and Max Welling. Semi-supervised classiﬁcation with graph convolutional networks. In ICLR 2017
>
> [6] Kuan Li, Yang Liu, Xiang Ao, Jianfeng Chi, Jinghua Feng, Hao Yang, and Qing He. Reliable representations make a stronger defender: Unsupervised structure reﬁnement for robust gnn. In KDD 2022.
>
> [7] Wei Jin, Yao Ma, Xiaorui Liu, Xian-Feng Tang, Suhang Wang, and Jiliang Tang. Graph structure learning for robust graph neural networks. In KDD 2020.
>
> [8] Wei Jin, Tyler Derr, Yiqi Wang, Yao Ma, Zitao Liu, and Jiliang Tang. Node similarity preserving graph convolutional networks. In WSDM 2021.
>
> [9] Felix Mujkanovic, Simon Geisler, Stephan Günnemann, and Aleksandar Bojchevski. Are defenses for graph neural networks robust? In NeurIPS 2022.

---

> ### Author Response · Authors · 2023-11-17
> **Followup on response to Reviewer knCN**
>
> Dear Reviewer knCN,
>
> We are respectfully writing to follow up on our previous response with regard to the review of our paper. We have carefully considered your feedback and provided detailed clarifications to address each of the issues raised, particularly those stemming from misunderstandings related to our methods, the adaptive attacks tested, and the causality of causal features. We believe that the additional information we've included in our response can clarify these points, which were initially presented in our paper.
>
> We eagerly await your feedback and are ready to offer any further clarifications or information you may require.
>
> Thank you once again for your recognition and contributions to our work.
>
> Best regards,
>
> The authors

---

> ### Author Response · Authors · 2023-11-21
> **Follow-up: Discussion deadline approaching**
>
> Dear Reviewer knCN,
>
> With the discussion deadline approaching, we wish to respectfully remind you of the pending feedback for our paper. While we have received responses from other reviewers to our clarifications, which have been greatly appreciated, we have not received your valuable input on our rebuttal and the additional materials we have submitted.
>
> Your insights are very important to us, and we are keen to hear your thoughts on our clarifications and revisions.  We eagerly await your feedback and are ready to offer any further clarifications or information you may require.
>
> Thank you once again for your recognition and contributions to our work.
>
> Best regards,
>
> The authors

---

> ### Author Response · Authors · 2023-11-22
> **Request for reply: Discussion deadline approaching**
>
> Dear Reviewer knCN,
>
> As the deadline approaches, we are writing to inquire about your reply to our clarification and revisions, as we have received replies from all the other reviewers. We wish to assure you that we are ready to address any queries or provide additional information promptly.
>
> We appreciate your recognition and contribution to our paper, and eagerly anticipate your valuable reply.
>
> Best regards,
>
> The authors

---

### Author Response · Authors · 2023-11-14
**Rebuttal by Authors**

Dear reviewers,

We would like to express our sincere thanks to all reviewers for your valuable feedback and for taking the time to evaluate our work. We are very encouraged by the recognition of our main contributions by all the reviewers. We have taken great care to address any concerns or misunderstandings raised by the reviewers by providing detailed clarifications and additional support. We hope our response can enhance the reviewers' confidence in our work. We believe that our efforts have been worthwhile and are looking forward to your continued attention to our paper.

To facilitate the assessment of our contributions, Let us summarize the strengths recognized by reviewers:

1. Novel motivation and perspective. (reviewer jN54,P55P, and knCN)
2. Interesting approach and effective method (reviewer P55P, jN54)
3. Sound experiments and significant improvement. (reviewer P55P, jN54, 6toz, knCN)
4. Complete theoretical derivation and proof. (reviewer jN54, 6toz)
5. Good presentation. (reviewer P55P)

We have noticed certain misunderstandings regarding our paper within the review comments, which we have addressed in our response. The key points of clarification are as follows:

1. Misunderstanding on scope of the attacks tested (reviewer jN54, knCN): We respectfully invite the reviewers to revisit the "Implementation" section detailed in Section 4 of our initial submission. Our experimental results broadly cover various types of 6 attacks, including poisoning/evasion attacks, untargeted/targeted attacks, graph modification/node injection attacks, as well as white-box adaptive attacks and gray-box attacks, highlighting a comprehensive evaluation of our method.
2. Misunderstanding on the role of causality (reviewer 6toz): We respectfully suggest the reviewer refer to Section 3.2.1 in our initial submission. In fact, the goals of IDEA are derived by leveraging the causality.  We also have provided a summary in our response to facilitate a rapid comprehension of causality's pivotal role in our method.
3. Concerns on conceptions (reviewer P55P): The term“causal” refers to features that maintain a direct and invariant causality with the labels, supported by lots of literature. Regarding “generalization on unseen attacks”,  IDEA aims to learn the causal features that determine labels across different attack domains, rather than spurious relationship limited to a single domain.

In sum, we believe our clarifications have effectively addressed the misunderstandings and concerns. We respectfully request that the reviewers re-evaluate our work based on our responses. We trust that our paper makes meaningful contributions to the field of graph adversarial learning and benefits the research community. Thank all the reviewers for your valuable and constructive comments on improving our paper.  If there are any further concerns that would prevent the reviewer from increasing the score, please let us know and we would be happy to address these concerns during the discussion phase.

Best regards,

The authors

---

### Author Response · Authors · 2023-11-23
**Summary of Author-Reviewer Discussions**

Dear Reviewers and Area Chairs,

We deeply appreciate the Reviewers for their insightful suggestions and the Area Chairs for their guidance throughout the review process. We are very encouraged that all reviewers have recognized our main contributions, including the novel motivation, interesting and effective method, sound experiments, as well as significance in the research field.

During the discussion, we provided clarifications and updated a revised paper to address each point raised by the reviewers. We believe our efforts have effectively addressed reviewers' concerns or misunderstandings, as evidenced by the improved evaluation scores from three reviewers (Reviewer P55P, Reviewer jN54, and Reviewer 6toz). Currently, the paper's scores are 5, 6, 6, and 8, with an average of 6.25.

To facilitate the assessment of our contributions, let us summarize the strengths recognized by reviewers:

1. Novel motivation and perspective. (reviewer jN54,P55P, and knCN)
2. Interesting approach and effective method (reviewer P55P, jN54)
3. Sound experiments and significant improvement. (reviewer P55P, jN54, 6toz, knCN)
4. Complete theoretical derivation and proof. (reviewer jN54, 6toz)
5. Good presentation. (reviewer P55P)

For a detailed account of the reviewers' concerns and our subsequent clarifications, please refer to the "Rebuttal by Authors". We are pleased to report that Reviewers P55P, jN54, and 6toz are satisfied with our responses and revisions. Unfortunately, we note that Reviewer knCN has not engaged in the discussion nor responded to our rebuttal.

We are hopeful that our paper will be favorably considered for presentation at the conference. We believe that our work offers a novel invariant causal defense paradigm that significantly improves graph adversarial robustness and advances the state-of-the-art, marking a promising new research direction with potential for future exploration.

Our sincere thanks go to all the Reviewers and Area Chairs for your invaluable contributions to the improvement of our paper.

Best regards,

The authors

---

### Meta-Review · Area_Chair_qCmc · 2023-12-08

**Metareview:**

The manuscript proposed a defense method to improve the graph adversarial robustness. At a high-level, the features are enforced to follow the two criterion: 1). they should be informative regarding the node classification task $Y$; 2). they should be invariant to the attacks/perturbation etc. From this perspective, I do think the term "invariance" should be better and more accurate to reflect the notion used in this paper rather than "causal". Technically, (conditional) mutual information is used to characterize the invariance criterion and gets further relaxed using its variational formulation. Such relaxation is standard in the literature, and has been widely used in existing work, as pointed out by both the authors themselves as well as the reviewers.

My main reservation for the paper in its current form is a lack of comparison with closely related works in the literature, both conceptually and experimentally. Conceptually, if we look at Eq. (3), the proposed method is very closely related to invariant risk minimization (over graphs) but the discussion is missing;  On the methodology level, the proposed loss approximation in Section 3.2.2 is almost the same as the one used in [1] (Section 5.2) and the reference is missing. Empirically, the authors also missed a relevant work [2] that is closely related to (in terms of problem setting) but different from (in terms of the design of the objective function) the proposed method.

[1].    Learning Invariant Representations and Risks for Semi-supervised Domain Adaptation
[2].    Information Obfuscation of Graph Neural Networks

I'd encourage the authors to further improve the paper by addressing the above concerns and incorporating the constructive feedback from the reviewers.

**Justification For Why Not Higher Score:**

Important related works are missing from the current version, both conceptually and empirically.

**Justification For Why Not Lower Score:**

N/A

---

### Decision · Program_Chairs · 2024-01-16

Reject